# Curious Exploration via Structured World Models Yields Zero-Shot Object Manipulation

**Cansu Sancaktar**          **Sebastian Blaes**          **Georg Martius**

Max Planck Institute for Intelligent Systems
Tübingen, Germany
{cansu.sancaktar, sebastian.blaes, georg.martius}@tue.mpg.de

## Abstract

It has been a long-standing dream to design artificial agents that explore their environment efficiently via intrinsic motivation, similar to how children perform curious free play. Despite recent advances in intrinsically motivated reinforcement learning (RL), sample-efficient exploration in object manipulation scenarios remains a significant challenge as most of the relevant information lies in the sparse agent-object and object-object interactions. In this paper, we propose to use structured world models to incorporate relational inductive biases in the control loop to achieve sample-efficient and interaction-rich exploration in compositional multi-object environments. By planning for future novelty inside structured world models, our method generates free-play behavior that starts to interact with objects early on and develops more complex behavior over time. Instead of using models only to compute intrinsic rewards, as commonly done, our method showcases that the self-reinforcing cycle between good models and good exploration also opens up another avenue: zero-shot generalization to downstream tasks via model-based planning. After the entirely intrinsic task-agnostic exploration phase, our method solves challenging downstream tasks such as stacking, flipping, pick & place, and throwing and generalizes to unseen numbers and arrangements of objects without any additional training.[1]

## 1 Introduction

Curious free-play has been identified as a driving force in child development, allowing children to efficiently explore their environment and build an understanding of the world [1]. Such intrinsically motivated exploration schemes are especially attractive in open-ended learning scenarios to guide an agent even without extrinsic tasks and corresponding rewards. Similar to how children learn, we want Reinforcement Learning (RL) agents to learn through play and then be able to solve new tasks quickly. We address both challenges in this work.

Minimization of novelty (or surprise) is a prominent formulation of curiosity, with several psychological studies showcasing the role of novelty in children's curious exploration [2–4]. Novelty as an intrinsic reward signal has also been adopted in RL, where agents try to resolve a cognitive disequilibrium [5, 6]. However, applying entirely intrinsic task-agnostic exploration to object manipulation scenarios in a sample-efficient manner is an ongoing challenge as the relevant information lies in the sparse agent-object and object-object interactions. These compositional multi-object manipulation environments highlight one of the significant weaknesses of the current novelty-based intrinsic motivation methods: a novel stimulus alone does not necessarily mean that it contains useful

---

[1]Code and videos are available at https://martius-lab.github.io/cee-us.

36th Conference on Neural Information Processing Systems (NeurIPS 2022).

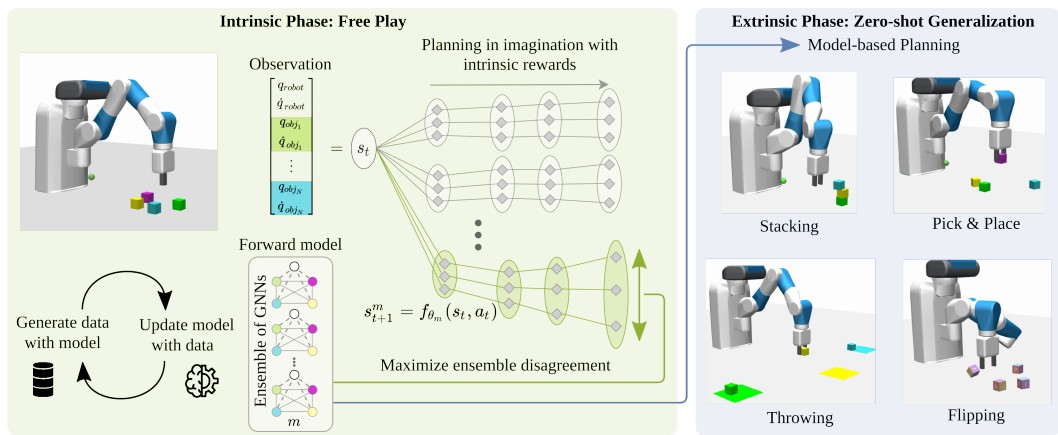

**Figure 1**: **Intrinsic motivation for epistemic uncertainty reduction with structured world models yields capable models for zero-shot task planning.** In an intrinsic phase, the system plans for maximum epistemic uncertainty with an ensemble of GNNs and uses the actively collected data to update the model. This iterative procedure ultimately reduces the overall epistemic uncertainty of the system, i.e. maximizing information gain. The learned models can be used to solve complex manipulation tasks using online planning.

or generalizable information for an individual [7]. Thus, curious exploration needs to attend to a subset of possibilities, as supported by studies in psychology [8, 9]. Analyzing children during free play accumulated evidence that infants have innate biases and heuristics to help guide their attention toward relevant and informative features of the environment [10, 11]. Our goal is to improve curious exploration in RL by incorporating appropriate inductive biases. We hypothesize that viewing the world as a collection of entities and their interactions is such a "useful" inductive bias, which was also put forward in Tsividis et al. [12]. However, instead of utilizing explicit theory-based modeling, we exploit a relational inductive bias by using Graph Neural Networks (GNN) as our choice of model [13] and learn everything from interactions. We postulate that curious exploration via epistemic uncertainty of such a model leads to the collection of valuable data as it inherits this structure.

This paper shows how online planning methods can efficiently exploit learned models, both for exploration and zero-shot generalization to tasks. Recent advances in general-purpose model predictive control methods have reduced their complexity and show strong performance when good forward models are available [14, 15]. We build on this work to create a self-reinforcing cycle between learning good models and good exploration. Thanks to their ability to plan for a new task without further training, models efficiently track the naturally changing exploration targets and perform downstream tasks in a zero-shot generalization manner.

We propose CEE-US: **C**urious **E**xploration using **E**pistemic **U**ncertainty via **S**tructured Models that achieves sample-efficient and interaction-rich exploration in multi-object manipulation environments. To our knowledge, we are the first to use GNN ensemble disagreement for computing intrinsic motivation signals in RL. A further major contribution of our method is to combine GNN-based epistemic uncertainty with planning methods in a task-agnostic setting and demonstrate zero-shot generalization to challenging object manipulation tasks. We provide a brief comparison to related work in Table 1 and provide more details in Sec. 4.

**Table 1**: **Comparison of CEE-US with baselines** in terms of differences in paradigms.

| Method | Observation type | Planner or policy | Learns dynamics | Zero-shot task generalization | Combinatorial generalization |
|---|---|---|---|---|---|
| CEE-US | Proprioceptive | Planner | ✓ | ✓ | ✓ |
| MLP + iCEM | Proprioceptive | Planner | ✓ | ✓ | ✗ |
| Disagreement [16] | Propr. + Image | Policy | ✓ | ✗ | ✗ |
| RND [17] | Propr. + Image | Policy | ✗ | ✗ | ✗ |
| ICM [6] | Propr. + Image | Policy | ✓ | ✗ | ✗ |
| Plan2Explore [18] | Image | Policy | ✓ | ✓(offline RL) | ✗ |

# 2 Method

We focus on intrinsically motivated learning to prepare for future tasks. Thus, we have a task-agnostic RL setting without extrinsic rewards or any other information regarding future tasks/goals during the initial play phase. Our approach trains a structured world model to capture the forward dynamics of the environment from active exploration data. In particular, we are using a Graph Neural Network (GNN) where the nodes correspond to objects. We rely on our model's epistemic uncertainty to direct exploration during free play. Our focus is on compositional multi-object environments, where an actuated agent can independently manipulate different objects. The terms (actuated) agent and robot are used interchangeably.

## 2.1 Preliminaries

In this work, we consider a fully observable Markov Decision Process (MDP) setting, The MDP is given by $\mathcal{M} = (\mathcal{S}, \mathcal{A}, P_{ss'}^a, R_{ss'}^a)$, with the continuous state-space $\mathcal{S} \in \mathbb{R}^{n_s}$, the continuous action-space $\mathcal{A} \in \mathbb{R}^{n_a}$, the transition kernel $P_{ss'}^a$, and the reward function $R_{ss'}^a$. Furthermore, we consider an object-oriented state representation, i.e. the state-space factorizes into the different entities in the environment $\mathcal{S} = \mathcal{S}_1 \times \mathcal{S}_2 \cdots \times \mathcal{S}_N \times \mathcal{S}_{\text{agent}}$, where $N$ denotes the number of objects.

In the typical RL setting, the agent's goal is to learn a policy $a \sim \pi(\cdot \mid s)$ that maximizes the future (discounted) cumulative reward $G_t = \sum_{k=0}^{\infty} \gamma^k \cdot R_{t+k}$, with the discount factor $0 < \gamma \leq 1$. Such a policy can be learned as a neural network with RL algorithms, or a planning method can be used to maximize the same quantity.

### 2.1.1 Planning and Model Predictive Control

Planning methods use a model $f$ of the transition kernel $P_{ss'}^a$ and a reward function $R_{ss'}^a$ to optimize a sequence of actions on the fly. The potential advantage of planning methods is that they can optimize for different reward functions without further adaptation. For this to be successful, a good transition model $f$ needs to be learned, and the reward function needs to be known or discovered. More formally, for a fixed planning horizon $H$, the action sequence $\mathbf{a}_t = (a_{t+h})_{h=0}^{H-1}$ is optimized to maximize

$$\mathbf{a}_t^{\star} = \arg\max_{\mathbf{a}_t} \sum_{h=0}^{H-1} R(s_{t+h}, a_{t+h}, s_{t+h+1}), \tag{1}$$

where $s_{t+h}$ are imagined states visited by rolling out the actions using $f$, which is assumed to be deterministic. For this procedure to optimize the same infinite horizon reward with discounting, $\gamma$ and a value function $V(s_{t+H})$ could be added to Eq. 1, sacrificing the flexibility in exchanging $R$.

We use zero-order trajectory optimization to find actions according to Eq. 1 and model predictive control (MPC) to convert the open-loop planning policy into a closed-loop policy (re-planning after every step in the environment). In particular, we use the improved Cross-Entropy Method (iCEM) [14], which was recently proposed as a sample efficient trajectory optimizer.

## 2.2 World Model with Graph Neural Networks

As the forward dynamics $f : \mathcal{S} \times \mathcal{A} \to \mathcal{S}$ (world model) is an integral part of the planning procedure, it needs to be able to capture the true dynamics well. We employ an ensemble of message-passing GNNs [13]. Each object corresponds to a node in the graph, and the node attributes $\{s_t^n \in \mathcal{S}_n \mid n = 1, \dots, N\}$ are given by an object's features such as position, orientation, and velocity at timestep $t$. The state representation of the actuated agent $s^{\text{agent}} \in \mathcal{S}_{\text{agent}}$ similarly contains position and velocity information about the robot.

The agent representation is differentiated from the object nodes as it has a direct cause-and-effect relationship with the actions. The agent's state and the generic action are represented as a global context $c$. The GNN is fully connected, i.e. there is an edge between all nodes. In GNNs, the node update function $g_{\text{node}}$ models the dynamics of individual entities, and the edge update function $g_{\text{edge}}$ captures their pairwise interactions.

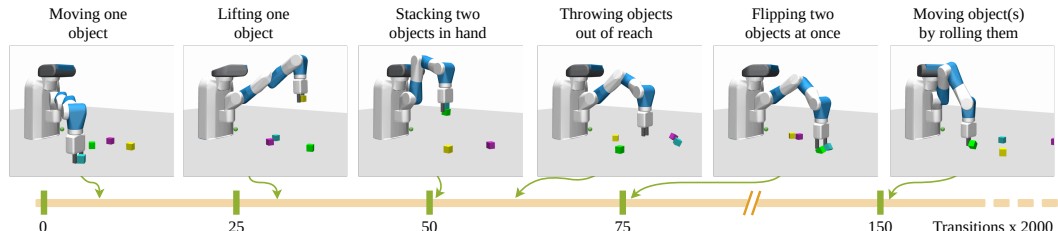

**Figure 2**: **Emergent behaviors observed during free play** with CEE-US in CONSTRUCTION. Along the time axis, 2000 transitions correspond to one training iteration of CEE-US, with 20 rollouts (episodes) of length 100. After only 30 iterations of free play, the robot starts lifting objects.

The node attribute computation for the objects is given by:

$$c = [s_t^{\text{agent}}, a_t] \tag{2}$$

$$e_t^{(i,j)} = g_{\text{edge}}\big([s_t^i, s_t^j, c]\big) \tag{3}$$

$$\hat{s}_{t+1}^i = g_{\text{node}}\big([s_t^i, c, \text{aggr}_{i \neq j}\big(e_t^{(i,j)}\big)]\big). \tag{4}$$

where $[\cdot, \ldots]$ denotes concatenation, $e_t^{(i,j)}$ is the edge attribute between two neighboring nodes $(i, j)$, and $c$ is the global context information. The permutation-invariant aggregation function is given by aggr. The $g_{\text{node}}$ and $g_{\text{edge}}$ are both Multilayer Perceptrons (MLP).

The context update, i.e. the transition of the agent's state, is computed using the global aggregation of all edges similar to the global context formalism in Battaglia et al. [13]:

$$\hat{s}_{t+1}^{\text{agent}} = g_{\text{global}}\big([c, \text{aggr}_{i,j}\big(e_t^{(i,j)}\big)]\big), \tag{5}$$

where $\mathbf{g}_{\text{global}}$ denotes the global node MLP. By providing all information to the prediction of the agent, we ensure an accurate modeling of agent-object interactions and their influence on the agent, which is paramount in object manipulation environments. Using this representation, we found that a single message passing step is sufficient, thus yielding fast inference times.

Moreover, to make the model focus on capturing the changes accurately, we let the GNN predict $\Delta\hat{s}_{t+1} = \hat{s}_{t+1} - s_t$, instead of the absolute next state, such that $f(s, a) = s + GNN(s, a)$

### 2.3 Epistemic Uncertainty as Intrinsic Reward

We train an ensemble of GNNs $\{(f_{\theta_m})_{m=1}^M\}$ (Fig. 1), where $M$ denotes the ensemble size. The epistemic uncertainty, i.e. the uncertainty due to lack of data, can be approximated by the disagreement of the ensemble members' predictions measured by the trace of the covariance matrix [15]:

$$R_{\text{I}}(s_t, a_t, \cdot) = \text{tr}\big(\text{Cov}(\{\hat{s}_{t+1}^m = f_{\theta_m}(s_t, a_t) \mid m = 1, \ldots, M\})\big). \tag{6}$$

**Connections to Information Gain** If more data becomes available from a region where the ensemble members disagree, then we expect the disagreement in this region to shrink, given enough capacity of the predictors. By guiding the exploration to regions with large model disagreement or high epistemic uncertainty, we explicitly aim for maximizing information gain [19–21]. Interestingly, as we can *predict* the uncertainty for unvisited states, our approach is more related to predicted information gain [22].

### 2.4 The CEE-US Algorithm

CEE-US has two different phases: the intrinsic free-play phase (Alg. 1), where learning occurs with active exploration, and the extrinsic phase (Alg. 2), where we apply the learned model to solve downstream tasks zero-shot without any additional training.

**Intrinsic Phase** In each iteration, the zero-order trajectory optimizer is used to plan for action sequences with high cumulative epistemic uncertainty in the imagined trajectories of the world model. The collected rollouts are added to the buffer, and the ensemble members of the world model are

trained on the observed state transitions $(\mathbf{s}_t, \mathbf{a}_t, \mathbf{s}_{t+1})$ for a fixed number of epochs minimizing the loss function:

$$\mathcal{L}_m = \|\Delta s_{t+1} - f_{\theta_m}(s_t, a_t)\|_2^2 \tag{7}$$

for each ensemble member $m$ with independently sampled mini-batches, where $\Delta s_{t+1} = s_{t+1} - s_t$ denotes the actual change in the next state observation. The total loss function is given by: $\mathcal{L} = \sum_{i=0}^{M} \mathcal{L}_m$. Afterwards, the process of data collection and training is repeated. The pseudocode for the intrinsic phase is provided in Alg. 1. Lines 3–9 correspond to one training iteration.

---

**Algorithm 1 CEE-US: Free Play in Intrinsic Phase**

1: **Input:** $\{(f_{\theta_m})_{m=1}^{M}\}$: Randomly initialized ensemble of GNNs with $M$ members, $D$: empty dataset, `Planner`: iCEM planner with horizon $H$
2: **while** explore **do**    ▷ *Explore with MPC and intrinsic reward*
3:    **for** $e = 1$ **to** num_episodes **do**
4:      **for** $t = 1$ **to** $T$ **do**    ▷ *Plan to maximize model uncertainty*
5:        $a_t \leftarrow$ `Planner`$(s_t, \{(f_{\theta_m})_{m=1}^{M}\}, R_{\mathrm{I}})$    ▷ *Eq. 6*
6:        $s_{t+1} \leftarrow$ `env.step`$(s_t, a_t)$
7:      $\mathcal{D} \leftarrow \mathcal{D} \cup \{(s_t, a_t, s_{t+1})_{t=1}^{T}\}$
8:    **for** $l = 1$ **to** $L$ **do**    ▷ *Train models on dataset for L epochs*
9:      $\theta_m \leftarrow$ optimize $\theta_m$ using $\mathcal{L}_m$ on $\mathcal{D}$ for $m = 1, \ldots, M$
10: **return** $\{(f_{\theta_m})_{m=1}^{M}\}, \mathcal{D}$

**Algorithm 2 CEE-US: Zero-shot Generalization in Extrinsic Phase**

1: **Input:** $\{(f_{\theta_m})_{m=1}^{M}\}$: Ensemble of GNNs with $m$ members learned during intrinsic phase, `Planner`: iCEM planner with horizon $H$, $R_{\mathrm{task}}$: Reward function of extrinsic task
2: **while not** done **do**    ▷ *Plan to maximize task reward in model*
3:    $a_t \leftarrow$ `Planner`$(s_t, \{(f_{\theta_m})_{m=1}^{M}\}, R_{\mathrm{task}})$    ▷ *task reward*
4:    $s_{t+1} \leftarrow$ `env.step`$(s_t, a_t)$

---

**Extrinsic Phase** The learned world model is used to perform planning for several downstream tasks, which we assume to be given in terms of reward functions $R_{\mathrm{task}}(s, a, s')$. The pseudocode is presented in Alg. 2. Here, no further adaptation to the model is performed, although our method could be extended to incorporate a potential fine-tuning procedure.

## 3 Experiments

In our empirical evaluation, we analyze the performance of CEE-US on two different object manipulation environments to answer the questions: How much does the structural inductive bias introduced by GNNs help model learning and control? Does the free-play phase create rich interaction data that helps downstream task performance? Can we solve challenging manipulation tasks in a zero-shot manner? The two environments we consider are:

**Playground** An actuated spherical agent can slide in the x-y directions and push four different object types (light cube, heavy cube, pyramid, and cylinder) along the x-y directions and also rotate them around the z-axis (Fig. 3). Each object has 3 Degrees of Freedom (DoF). Object types are uniquely identified by their color.

**Fetch Pick & Place Construction** This is an extension of the Fetch Pick & Place environment [23] to more cubes [24] (Fig. 2). A 7-DoF robot arm is used to manipulate blocks (each with 6 DoF). The actions $a \in \mathbb{R}^4$ control the gripper movement in Cartesian coordinates and the gripper opening/closing. The robot state $\mathbf{s}_{\mathrm{agent}} \in \mathbb{R}^{10}$ contains the end-effector position and velocity and the gripper-state (open/close) and gripper-velocity. Each object's state $s^i \in \mathbb{R}^{12}$ is given by its position, orientation (in Euler angles), and linear and angular velocities. We replaced the table in front of the robot with a large plane so that objects cannot fall off during free play. However, they can still be pushed or thrown outside of the manipulability range of the robot. Originally, each object state contained the object's position relative to the gripper. We remove this privileged information from the objects' state, as it already introduces a relational bias in the raw state representation.

### 3.1 Structured vs. Unstructured World Models

We analyze how curious exploration based on minimizing epistemic uncertainty performs when using ensembles of GNNs and ensembles of MLPs (Fig. 3). MLPs, with their fully connected layers and monolithic input representations, do not offer isolation of information and incorporate no explicit relational inductive bias [13], thus constituting a good baseline for our structured world models. Results in PLAYGROUND show that the GNN ensemble leads much faster to interaction-rich data

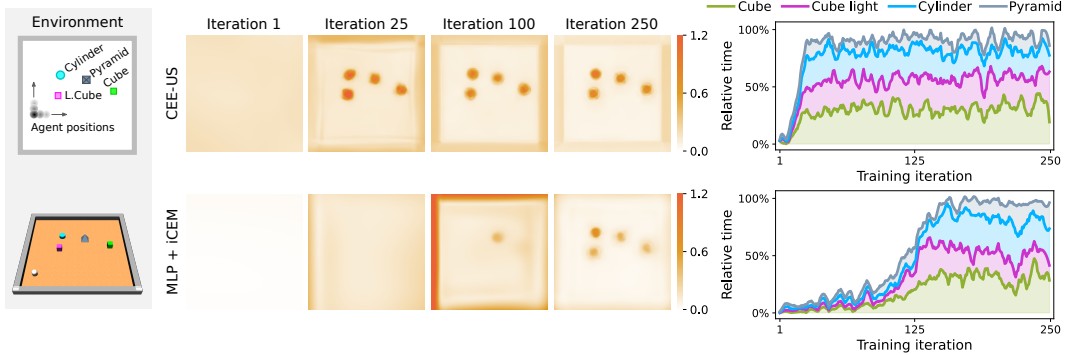

**Figure 3**: **Structured world models lead to more object interactions.** In the playground environment, (left) we compare active exploration with an ensemble of GNNs (top row) vs. an ensemble of MLPs (bottom row) using epistemic uncertainty as an intrinsic reward ($M = 5$). Middle: The heatmaps show the uncertainty of the ensemble members for hypothetical agent positions over a spatially discretized grid at different training iterations (see Suppl. C.6). Right: The relative time (the fraction of time steps) the agent spends moving the different objects in the playground is illustrated throughout free play.

than its MLP counterpart. As visualized in Fig. 3, the uncertainty produced by the GNN ensemble is already localized around objects after 25 training iterations, leading to targeted agent-object and object-object interactions. For the MLP, it takes more than 100 iterations to start generating useful uncertainty estimates and therefore object-agent interactions. Also, note the pronounced uncertainty at the walls for the MLP. A more fine-grained analysis of the interaction times of CEE-US compared to the baselines will be provided below. In Suppl. D, we also show how the resulting multi-step dynamics predictions with structured world models are more accurate compared to the MLPs. This is a key component for the self-reinforcing cycle between good models and good exploration as a better model means: the agent can plan for more complex behavior earlier and learn from these experiences faster.

### 3.2 Interaction-Richness of Generated Exploration Data

We perform an analysis of the data generated during intrinsic free-play. We compare the performance of our method to the unstructured MLP + iCEM variant (CEE-US without GNNs) as well as other intrinsic motivation baselines:

**Disagreement [16]**    The one-step disagreement of an MLP ensemble is used as an intrinsic reward to train an exploration policy.

**Random Network Distillation [17]**    A predictor network, corresponding to the forward model, tries to match the output of a target network with random weights. The discrepancy between the two networks is used as an intrinsic reward (a type of state visitation count for continuous domains).

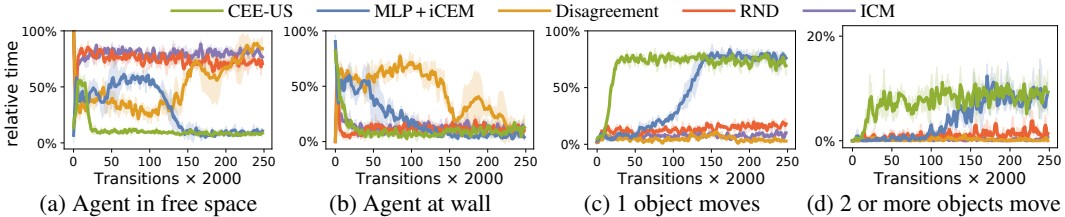

(a) Agent in free space    (b) Agent at wall    (c) 1 object moves    (d) 2 or more objects move

**Figure 4**: **Interaction rate during intrinsic exploration in the PLAYGROUND environment.** The relative interaction times show that CEE-US quickly starts interacting with the objects. MLP + iCEM is similar but one order of magnitude less data efficient. The baselines rarely engage in interesting interactions, mostly staying in free space or at the walls (three independent seeds).

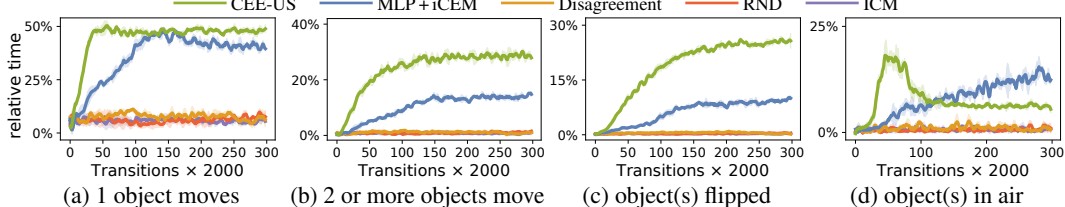

| (a) 1 object moves | (b) 2 or more objects move | (c) object(s) flipped | (d) object(s) in air |

**Figure 5**: **Rich interactions during free play in CONSTRUCTION by CEE-US**. Interaction metrics of free-play exploration count the relative amount of time steps spent in moving one object (a), moving two and more objects (b), flipping object(s) (c), and moving objects into the air (d). CEE-US engages quickly in all sorts of interactions with objects. The ablation with MLPs eventually performs similarly but is less sample efficient. The RL baselines perform poorly on these timescales. We used three independent seeds.

**ICM [6]** The intrinsic reward is defined as the error between an MLP forward model's next state prediction and the actual next state. As this method needs access to the true next state, the intrinsic reward can only be computed retrospectively.

The intrinsic reward is used to train an exploration policy in these methods. We use the implementation from Laskin et al. [25] that uses DDPG [26]. In Fig. 4, we present different metrics quantifying the amount and type of interactions that occur during the free play in the PLAYGROUND environment. It is eminent that CEE-US is only shortly interacting with the walls and then quickly interacts with one and two objects simultaneously. The next best method is the ablation of our method using MLP models needing about ten times more interactions, as explained in Fig. 3. The baselines are mainly moving in free space and interacting with the walls. RND starts interacting with the objects in at least 10% of the times-steps.

In the CONSTRUCTION environment, the situation is even more drastic, as shown in Fig. 5. CEE-US starts repeatedly moving one object after only eight iterations (each 2000 environment steps), picks up objects after around 30 iterations, and continues with throwing and flipping objects and moving multiple objects frequently after 50 iterations. In Fig. 2 exemplary behaviors are shown in terms of snapshots at different stages of learning. We believe this sample efficiency is remarkable. The ablation of our method with MLP models is also able to engage in interesting and diverse interactions, but at a much slower rate (Fig. 5). The policy learning baselines rarely interact with objects, even after 300 iterations with 20 episodes each.

## 3.3 Zero-Shot Generalization to Downstream Tasks

We demonstrate that after the free-play phase, the learned models can be used for zero-shot solving of complex downstream tasks without the necessity to generate new data or to perform further training. Furthermore, thanks to the combinatorial generalization capabilities of GNNs, the model can be used with a different number of objects in the environment than seen during free play. The only baseline we consider for the zero-shot generalization performance is MLP+iCEM, as none of the other methods discussed in the previous section can solve downstream tasks without additional training.

In PLAYGROUND, we consider the task of bringing all objects to a single target location, as shown in Fig. 6. The reward function is the sum of negative distances of all objects to the target outside a threshold distance. The target location is sampled randomly at the beginning of each episode. The

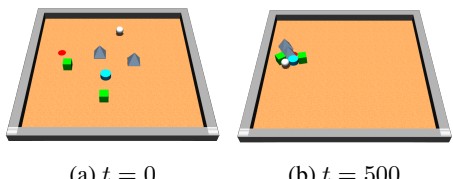

| (a) $t = 0$ | (b) $t = 500$ |

**Figure 6**: Downstream task in PLAY-GROUND: move all objects to one place, as solved by CEE-US.

**Table 2**: **Success rates in the PLAYGROUND environment** for the downstream task *Push objects to one location*, with a variable number of objects after 100 and 250 training iterations of free play. Computed over three random seeds, each evaluated on 100 roll-outs.

| | 4 Objs@100 one each | 4 Objs@250 one each | 3 Objs@250 random | 5 Objs@250 random |
|---|---|---|---|---|
| CEE-US | $0.70 \pm 0.04$ | $0.91 \pm 0.06$ | $0.93 \pm 0.03$ | $0.93 \pm 0.04$ |
| MLP+iCEM | $0.08 \pm 0.01$ | $0.90 \pm 0.03$ | — | — |

success rates for different numbers of objects are shown in Table 2. We define success rate in the multi-object setup as the fraction of objects solved relative to the total number of objects spawned in the environment. After 100 training iterations of free play, CEE-US already achieves a 70% success rate, whereas the MLP version only reaches 8%. After 250 iterations, both methods are on par. Since CEE-US can also deal with a variable input dimension, we also consider the task with more or less objects with randomly sampled types.

In CONSTRUCTION, several challenging manipulation tasks need to be solved: pick & place, stacking, throwing, and flipping, as shown in Fig. 1. The reward functions are detailed in Suppl. C.3.1. In Fig. 7, we present the success rates in each of the tasks for CEE-US and the MLP-based planning baseline MLP + iCEM, where applicable. We find remarkable success rates across the board for CEE-US, even in challenging tasks such as stacking and throwing. For stacking, we report a success rate of 1 if the tower with all the objects in the environment is stacked and 0 otherwise. For the multi-tower task, denoted by $2 + 2$, we consider success 1 only when two towers are stacked.

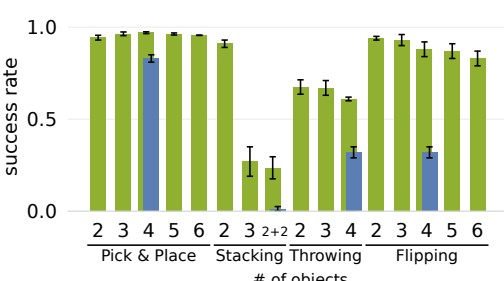

Figure 7: **Zero-shot performance on downstream tasks in CONSTRUCTION** for CEE-US and MLP + iCEM.

We recommend visiting our website[2] for videos of these tasks. In Suppl. E, we present additional experiments combining RND as intrinsic reward with model-based planning, similar to Lambert et al. [27], in CONSTRUCTION, where we look at both structured and unstructured world models. The results showcase the benefits of using the model's own epistemic uncertainty estimate to guide exploration, as it is the case with ensemble disagreement, leading to more accurate and more robust dynamics models and better zero-shot downstream task generalization.

### 3.4 Offline Learning of Downstream Tasks from Exploratory Data

This experiment investigates the quality of the data collected by the different methods in the free-play phase for solving downstream tasks via offline RL. This allows us to compare to other baselines that do not support zero-shot generalization. Additionally, it is one way of obtaining task-specific policies. In offline RL, a control policy is learned from a fixed dataset, where we repurpose the exploration data. The results are shown in Table 3. Offline policy learning is performed via CQL [28][3] and training details can be found in Suppl. C.5. To learn different downstream tasks from the same dataset, reward relabeling and hindsight experience replay [30] are used.

In accordance with the interaction metrics during the intrinsic phase (Figs. 4,5), CEE-US and MLP + iCEM achieve the highest performance in all four tasks, supporting the benefit of optimizing for long-horizon future novelty. Especially in the object manipulation tasks, data from CEE-US shows a clear advantage. Tasks with more objects could not be solved with the amount of data collected.

**Table 3**: **Success rates on tasks with offline RL** trained from exploration data generated during the intrinsic phase with the respective method. Using CQL and HER on the data for each task, we see that CEE-US creates much more useful data for offline learning.

| Domain | Task | Disagreement [16] | RND [17] | ICM [6] | MLP + iCEM | CEE-US |
|---|---|---|---|---|---|---|
| PLAYGROUND | Locomotion | $0.05 \pm 0.01$ | $0.35 \pm 0.05$ | $0.01 \pm 0.0$ | $1.0 \pm 0.0$ | $1.0 \pm 0.0$ |
| 500k datapoints | Relocate 1 obj. | $0.06 \pm 0.01$ | $0.06 \pm 0.0$ | $0.06 \pm 0.01$ | $0.2 \pm 0.06$ | $0.9 \pm 0.01$ |
| CONSTRUCTION | Reach | $0.09 \pm 0.01$ | $0.19 \pm 0.05$ | $0.2 \pm 0.03$ | $0.65 \pm 0.09$ | $0.94 \pm 0.04$ |
| 600k datapoints | Pick & Place 1 obj. | $0.07 \pm 0.0$ | $0.07 \pm 0.0$ | $0.07 \pm 0.01$ | $0.18 \pm 0.06$ | $0.43 \pm 0.07$ |

## 4 Related Work

**Intrinsic motivation in RL** Prediction error [5, 6, 31], novelty and Bayesian surprise [20, 32, 33], learning progress [5, 32, 34], empowerment [35, 36] and count-based metrics [17, 27] are among the popular intrinsic reward signal definitions used in RL. These intrinsic rewards are either used to aid

---

[2]https://martius-lab.github.io/cee-us
[3]We are using the implementation from the *d3rlpy* library [29]

exploration in challenging tasks with sparse task rewards or in a task-agnostic setup where they are the only rewards. Algorithms using the task-agnostic setting follow two routes to solve downstream tasks: (i) they re-label the collected data during intrinsic exploration with the downstream task reward and perform offline RL [18, 27, 37] or (ii) they use snapshots of the exploration policy for bootstrapping and fine-tune them on downstream tasks [25, 38]. The first variant does not require additional interaction with the environment, however, additional training of a new policy is still necessary. In addition, offline RL struggles with the distribution shift, while we show that a good world model can generalize better. The second variant suffers from another inherent issue: the emerging behaviors of the exploration policy are lost during training. Although Groth et al. [38] address this problem by snapshotting and performing hierarchical RL, this is more of a bandaid solution. Many intrinsically motivated learning systems use a goal-achieving setting with predefined goal spaces [32, 34, 39, 40] or auxiliary tasks [41]. The differences in the existing paradigms are highlighted in Table 1. Methods such as ICM [6] rely on retrospective intrinsic motivation (here prediction error), i.e. the agent has to already be in the next state to assess how novel that state is. In order to accommodate planning for multi-step intrinsic motivation signals into the future, we need a way to predict them. Haber et al. [42], for instance, do so by learning a loss model to predict the prediction error at future time steps. Another line of research, including our method, uses the disagreement of an ensemble of world models to estimate the predictive information gain as the intrinsic motivation measure [16, 18]. Plan2Explore [18] shares similarities to our approach in the task-agnostic exploration phase, using multi-step ensemble disagreement as intrinsic reward. It works with a latent dynamics prediction model and has been applied to domains with image observations. However, Plan2Explore does not use structured world models and lacks mechanisms to achieve combinatorial generalization that is beneficial for sample-efficient exploration in object manipulation tasks.

**Relational Networks**  Several works showcase improved dynamics prediction performance in environments with interacting entities using structured world models [43–45]. Kipf et al. [43] uses a GNN to learn the latent transition dynamics in simple manipulation tasks with 2D shapes and 3D blocks from images. However, they use an object-factorized action space and do not tackle exploration but rely on an external dataset. Sanchez-Gonzalez et al. [44] applies GNNs to learn the dynamics of physical bodies, where the entities correspond to joints, also without active exploration. Driess et al. [46] uses GNNs and NERFs to obtain object-centric representations from images and RRT planners. Biza et al. [47] also utilizes the combinatorial generalization of structured world models to achieve zero-shot task generalization. However, the policy uses parameterized high-level actions and the world model is learned on an offline expert dataset, thus sidestepping the exploration challenge. Outside the model-based paradigm, Li et al. [24] achieves block stacking in CONSTRUCTION using a GNN policy, attention and tailored learning curriculum. In contrast, we use GNNs for computing intrinsic rewards and online planning.

**Curiosity in Object-Centric RL**  Watters et al. [48] deploy curious exploration with retrospective prediction error as an intrinsic reward in an object-centric setting with image inputs and a pre-trained vision model for object discovery, but without object-object interactions. In Seitzer et al. [49], an object-centric causal action-influence is used as an intrinsic reward to improve sample efficiency.

## 5   Discussion

In this work, we present CEE-US that combines the learning of GNNs as structured world models with curiosity-driven, planning-based exploration. We tackle a significant challenge that is not often addressed in existing intrinsically motivated RL works: guiding exploration towards potentially **useful** components of the environment. By approximating information gain via world models injected with relational inductive biases, CEE-US focuses on the interactions between entities in the environment. Maximizing future information gain via multistep forward-planning enables CEE-US to discover interaction-rich behaviors more efficiently than the exploration-policy-based baselines, as well as planning-based paradigms without structured world models.

After the intrinsic free-play phase, we use the learned GNNs to solve downstream tasks with model-based planning and zero-shot without any additional training. CEE-US can achieve zero-shot generalization without the need for an additional policy learning step, a strength of CEE-US that is missing from all the baselines. We show that even non-structured MLPs achieve zero-shot generalization on some tasks, albeit with lower success rates. This indicates that there are benefits

to learning good world models and utilizing them for control even without any further structural biases. This also ties into knowledge-based intrinsic motivation, as the experience gathered during free play is distilled and stored in the learned world models instead of being dismissed. It is important to note that using model-based planning during the intrinsic phase does not restrict us to model-based approaches in the extrinsic phase. The learned model could also be used to extract a task policy later using model-based policy optimization [50] or to perform offline RL on the data generated during free play, which we demonstrate in Sec. 3.4.

Despite the sample efficiency, there are some limitations to model-based online planning. The complexity of behaviors discovered in free play is upper-bounded by the finite planning horizon. The same also applies to the extrinsic phase, where solving tasks like throwing objects or solving multistep manipulation tasks require longer planning horizons without extensive reward shaping.

Although CEE-US currently uses proprioceptive state information, we do not assume access to any privileged information such that there are no fundamental limitations prohibiting us from applying it to real robots. Following Kipf et al. [43] and Watters et al. [48], CEE-US can also be extended to deal with image inputs and use methods like SCALOR [51] to extract object-centric unsupervised representations. The differentiation between agent and objects can be identified unsupervised [52].

The demonstrated sample efficiency in the unsupervised learning of capable world models opens new avenues for learning directly on real hardware. To put the training time into context, 45 training iterations correspond to 1h of interaction time. So our downstream performance in CONSTRUCTION was achieved after about 6.5h of free-play from scratch.

## Acknowledgments and Disclosure of Funding

The authors thank Arash Tavakoli and Pavel Kolev for helpful discussions and Andrii Zadaianchuk, Christian Gumbsch and Nico Gürtler for their help reviewing the manuscript. The authors thank the International Max Planck Research School for Intelligent Systems (IMPRS-IS) for supporting Cansu Sancaktar and Sebastian Blaes. Georg Martius is a member of the Machine Learning Cluster of Excellence, EXC number 2064/1 – Project number 390727645. We acknowledge the financial support from the German Federal Ministry of Education and Research (BMBF) through the Tübingen AI Center (FKZ: 01IS18039B). This work was supported by the Volkswagen Stiftung (No 98 571).

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
