# Supplementary Material for
# Curious Exploration via Structured World Models Yields Zero-Shot Object Manipulation

## A  GNN Architectural Details

We use message-passing GNNs in CEE-US as described in Sec. 2.2. When different object types are present in the environment, we also include static object features in the object states $s_t^i$. The dynamic component $s_t^{\text{dyn},i}$ is time-dependent and contains object positions and velocities, whereas the time-independent static features $s^{\text{stat},i}$ contain identifiers for different object types. For the overall object state at time step $t$, we get $s_t^i = [s_t^{\text{dyn},i}, s^{\text{stat},i}]$. This can be viewed as a concatenation of a dynamic and a static graph [44]. The GNN only makes a next state prediction for the dynamic state component of the object nodes so that e.g. the node update is given by:

$$\hat{s}_{t+1}^{\text{dyn},i} = g_{\text{node}}\big(\big[s_t^{\text{dyn},i}, s^{\text{stat},i}, c, \text{aggr}_{i \neq j}\big(e_t^{(i,j)}\big)\big]\big). \tag{S1}$$

We use static features only in the PLAYGROUND environment, where we have 4 different object types (cube, light cube, cylinder and pyramid) and the color of each object is used as the static feature. The mean is used as the permutation-invariant aggregation function aggr.

## B  Planning Details

For planning, we use the improved Cross-Entropy Method (iCEM) [14]. The pseudocode is given in Alg. S1. The costs for the planner correspond to negative reward such that:

$$C(s_t, a_t, s_{t+1}) = -R(s_t, a_t, s_{t+1}),$$

where $R$ can be intrinsic rewards $R_I$ or extrinsic task rewards $R_{\text{task}}$. Whenever we are dealing with an ensemble of models, we use the same notation $R(s_t, a_t, s_{t+1})$, even though the reward function in this case takes the transitions of the whole ensemble $\{(s_t^m, a_t, s_{t+1}^m)\}_{m=1}^M$ as input arguments. [1]

The algorithm shown here is for a single model $f_\theta$. In the case of an ensemble of models $\{(f_{\theta_m})_{m=1}^M\}$ with ensemble size $M$, each model sees the same P sampled action trajectories with each $\mathbf{a}_t = (a_{t+h})_{h=0}^{H-1} \in \mathbb{R}^{n_a \times H}$. During the intrinsic phase of CEE-US, the intrinsic rewards for planning are computed based on the ensemble disagreement, such that $R_I(s_t, a_t, s_{t+1})$ for one time step in a simulated trajectory is a scalar, computed according to Eq. 6.

In the extrinsic phase when we have task-specific reward functions, we utilize the different ensemble predictions for more robust action selection. Each model of the ensemble creates a cost trajectory for each sampled action sequence with $\{R_{\text{task}}(s_{t+h}, a_{t+h}, s_{t+h+1})\}_{h=0}^{H-1}$ such that the overall cost of a sampled trajectory amounts to a tensor with size $M \times H$. In order to then select the elites, we average the costs over the ensembles.

In a generalized setup, the sum in Eq. 1 can be replaced with another permutation-invariant function $\phi$. For planning, other than the default mode `sum` as shown in equation Eq. 1, we also allow mode `best` with $\phi = \max\big(\{R(s_{t+h}, a_{t+h}, s_{t+h+1})\}_{h=0}^{H-1}\big)$, which chooses the optimal trajectory according to the best reward observed at any time step over the planning horizon.

## C  Experiment Details

In this section, we provide experimental details and hyperparameter settings.

---

[1]Note that we overload the superscript to both indicate ensemble members' predictions and object-centric state representations. The index $m$ is for the the prediction of ensemble member $m$ on the whole state, and $i$ signals that we are looking at the state of object $i$.

**Algorithm S1** Model predictive control with iCEM planner

---

1: **Input:** $P$: number of samples; $H$: planning horizon; $n_a$: action dimension; $K$: size of elite-set; $\beta$: colored-noise exponent, *CEM-iterations*: number of iterations; $\xi$: fraction of elites reused; $\sigma_{init}$: noise strength, $\alpha$: momentum, $f_\theta$: Forward dynamics model, $\phi$: permutation-invariant function to compute overall cost over planning horizon.

2: **for** t = 0 **to** T−1 **do**  ▷ loop over episode length

3:     **if** t == 0 **then**

4:         $\mu_0 \leftarrow$ constant vector in $\mathbb{R}^{n_a \times H}$

5:     **else if** `shift_elites` **then**

6:         $\mu_t \leftarrow$ shifted $\mu_{t-1}$ (and repeat last time-step)

7:     $\sigma_t \leftarrow$ constant vector in $\mathbb{R}^{n_a \times H}$ with values $\sigma_{init}$

8:     **for** i = 0 **to** CEM-iterations−1 **do**

9:         samples $\leftarrow P$ samples from clip$(\mu_t + \mathcal{C}^\beta(n_a, H) \odot \sigma_t^2)$

10:        **if** i == 0 **and** `shift_elites` **then**

11:            add fraction $\xi$ of **shifted** elite-set$_{t-1}$ to samples

12:        **else if** `keep_elites` **then**

13:            add fraction $\xi$ of elite-set$_t$ to samples

14:        **if** i == last-iter **then**

15:            add mean to samples

16:        **for** h = 0 **to** H-1 **do**  ▷ Compute state trajectories in model $f_\theta$'s imagination

17:            $s_{t+h+1} \leftarrow f_\theta(s_{t+h}, a_{t+h})$ for $a_{t+h}$ in samples

18:        rewards $\leftarrow$ rewards of sampled trajectories $\phi(\{R(s_{t+h}, a_{t+h}, s_{t+h+1})\}_{h=0}^{H-1})$

19:        elite-set$_t \leftarrow$ best $K$ samples according to rewards

20:        $\mu_t, \sigma_t \leftarrow$ fit Gaussian distribution to elite-set$_t$ with momentum $\alpha$

21:    **if** `use_mean_actions` **then**

22:        execute first action of mean of elite sequences

23:    **else**

24:        execute first action of best elite sequence

---

## C.1 Intrinsic Phase with CEE-US

In the intrinsic phase of CEE-US, we iteratively generate rollouts with the iCEM planner using intrinsic rewards and then train the models of the ensemble on the overall data collected so far. We test CEE-US on the PLAYGROUND and CONSTRUCTION environments. The environment properties as well as the episode lengths and model training frequencies are given in Table S1. Four objects are present in each environment during free play (in PLAYGROUND: one of each object type). The parameters for the GNN model architecture as well as the training parameters for model learning are listed in Table S2. Note that model learning only occurs during the intrinsic phase. For the extrinsic phase, we take the learned model with the listed architectural settings to solve downstream tasks zero-shot.

The intrinsic free-play in CEE-US, together with the data collection and consequent model updates, is run for 300 training iterations in CONSTRUCTION, which takes roughly 72 hours using a single GPU (here NVIDIA GeForce RTX 3060) and 6 cores on an AMD Ryzen 9 5900X Processor. In PLAYGROUND, we run CEE-US for 250 training iterations which takes ca. 50 hours. Note that the duration of one training iteration increases throughout free-play, since we train the model for a fixed number of epochs on the whole data collected so far. As the number of transitions stored in the buffer increases, the number of update steps for the same number of epochs also increases.

## C.2 Controller Parameters

The set of default hyperparameters used for the iCEM controller are presented in Table S3, as well as environment-specific controller settings used for the intrinsic phase of CEE-US.

**Table S1**: Environment settings. In both environments 2000 transitions are generated within one training iteration of CEE-US.

| PLAYGROUND | | | CONSTRUCTION | |
|---|---|---|---|---|
| **Parameter** | **Value** | | **Parameter** | **Value** |
| Episode Length | 200 | | Episode Length | 100 |
| Train Model Every | 10 Episodes | | Train Model Every | 20 Episodes |
| Action Dim. | 2 | | Action Dim. | 4 |
| Robot/Agent State Dim. | 4 | | Robot/Agent State Dim. | 10 |
| Object Dynamic State Dim. | 6 | | Object Dynamic State Dim. | 12 |
| Object Static State Dim. | 3 | | Object Static State Dim. | 0 |

**Table S2**: Base settings for GNN model training in intrinsic phase of CEE-US.

(a) General settings.

| Parameter | Value |
|---|---|
| Network Size of $g_{\text{node}}$ | $2 \times 128$ |
| Network Size of $g_{\text{edge}}$ | $2 \times 128$ |
| Network Size of $g_{\text{global}}$ | $2 \times 128$ |
| Activation function | ReLU |
| Layer Normalization | Yes |
| Number of Message-Passing | 1 |
| Ensemble Size | 5 |
| Optimizer | ADAM |
| Batch Size | 125 |
| Epochs | 25 |
| Learning Rate | $10^{-5}$ |
| Weight Decay | 0.001 |
| Weight Initialization | Truncated Normal |
| Normalize Input | Yes |
| Normalize Output | Yes |
| Predict Delta | Yes |

(b) Environment-specific settings.

| PLAYGROUND | |
|---|---|
| **Parameter** | **Value** |
| Network Size of $g_{\text{node}}$ | $2 \times 64$ |
| Network Size of $g_{\text{edge}}$ | $2 \times 64$ |
| Network Size of $g_{\text{global}}$ | $2 \times 64$ |
| Learning Rate | 0.0001 |
| Weight decay | $5 \cdot 10^{-5}$ |

| CONSTRUCTION | |
|---|---|
| **Parameter** | **Value** |
| Same as general settings | |

## C.3 Extrinsic Phase

In this section, we provide details on the extrinsic phase of CEE-US, where the learned GNN ensemble is used to solve downstream tasks zero-shot via model-based planning.

### C.3.1 Details on Downstream Tasks and Reward Functions

We use the notation introduced in Sec. 2.1, where $s^i$ for $i = 1, \ldots, N$ denotes the state of each of the $N$ objects present in the environment and $g^i$ denotes the goal for each object in the environment. The superscript $i$ is omitted for the goal if all objects' goals are the same. For ease of notation in the reward function definitions, we consider $s^i$ to be the achieved goal state, which for all tasks other than flipping corresponds to the positional information of each object's state (x-y for PLAYGROUND and x-y-z for CONSTRUCTION). The actuated agent, i.e. robot, state is given by $s^{\text{agent}}$. Unless stated otherwise, the L2-norm is used to compute the distance between the current state $s$ and a target/goal state $g$ denoted by $\text{dist}(s, g) = \|s - g\|_2$. We use $\delta$ to denote the environment threshold for goal distances used to compute sparse rewards as well as potential cut-off values for dense rewards. The sizes in both environments are on different scales, so the used $\delta$ values vary. In PLAYGROUND, the spherical agent's diameter is $0.2$ and objects have size ca. $0.2$ with slight variations. In CONSTRUCTION, each cube/block has size $0.05$.

**PLAYGROUND-Pushing** The task in the PLAYGROUND environment is defined as bringing all objects to a target location $g \in \mathbb{R}^2$ that is sampled randomly in the beginning of each episode. The reward is defined as the sum of the negative distances of each object to the target location up to a

**Table S3**: Base settings for iCEM in CEE-US as well as the environment-specific settings used in the intrinsic phase. Same settings are used for MLP + iCEM.

(a) General settings.

| Parameter | Value |
|---|---|
| Number of samples $P$ | 128 |
| Horizon $H$ | 30 |
| Size of elite-set $K$ | 10 |
| Colored-noise exponent $\beta$ | 3.5 |
| *CEM-iterations* | 3 |
| Noise strength $\sigma_{\text{init}}$ | 0.5 |
| Momentum $\alpha$ | 0.1 |
| `use_mean_actions` | Yes |
| `shift_elites` | Yes |
| `keep_elites` | Yes |
| Fraction of elites reused $\xi$ | 0.3 |
| Cost along trajectory | `sum` |

(b) Environment-specific settings.

PLAYGROUND
*Intrinsic Phase*

| Parameter | Value |
|---|---|
| `shift_elites` | No |
| `keep_elites` | No |
| Noise strength $\sigma_{\text{init}}$ | 0.8 |

CONSTRUCTION
*Intrinsic Phase*

| Parameter | Value |
|---|---|
| Same as general settings | |

threshold distance $\delta$ such that:

$$R_{\text{push}} = \sum_{i=1}^{N} - \max(\text{dist}(s^i, g), \delta). \tag{S2}$$

The reason we have a cut-off at distance $\delta$ is to ensure that the agent doesn't unnecessarily try to bring each object to the exact center since the task is to bring all objects to the target location and the overall reward cannot be zero in the end with more than one object in the environment. We still use a small $\delta$ of 0.23 for this buffer zone in the experiments, such that the model still has to find a plan that focuses on the next unsolved object instead of optimizing for small gains with an object that is essentially already at the target $g$. Note that for evaluating whether the task was successfully solved, we define the distance threshold to be larger $\delta_{\text{eval}} = 0.35$, as to accurately account for cases where the target is in the corners or at the wall, such that not all objects can fit in the buffer zone area with the conservative $\delta$.

**CONSTRUCTION-Stacking**    For stacking, sparse incremental rewards are used with reward shaping. The shaped reward contains the (dense) distance between the gripper and the position of the next block to be stacked in the tower given by $s^{\text{next}}$. If the tower is fully stacked, then the shaped reward component $s^{\text{next}}$ contains the distance of the gripper to a resting position away from the tower base. In the experiments, we use $(0, 0, 0)$ as the goal position for the robot after it has finished stacking. The reward function is given by:

$$R_{\text{stack}} = \sum_{i=1}^{N} \left( -1 + [\![\text{dist}(s^i, g^i) < \delta]\!] \right) - \eta \cdot \text{dist}(s^{\text{agent}}, s^{\text{next}}), \tag{S3}$$

where $[\![\cdot]\!]$ are Iverson brackets and $\eta$ is the scale of the shaped reward component. In the experiments, we use $\delta = 0.02$ (each block has size 0.05) and $\eta = 0.01$. Note that in the original environment proposed in Li et al. [24], the distance threshold for all tasks was defined to be 0.05. However, in order to ensure stable stacking we reduce this to 0.02 and also use this same value for evaluating for a successful stack. We do not allow any mismatch between $\delta$ and $\delta_{\text{eval}}$, when we have sparse rewards.

**CONSTRUCTION-Pick & Place**    The task is defined as bringing each object $i$ to its individual goal position $g_i$ that is randomly sampled. In an environment with $N$ objects, the first $N - 1$ have goal positions on the ground. The $N$th object's goal is in air with 50% probability, where the target height is also sampled randomly. We use a dense reward with the sum of the negative distances of each object position $s^i$ to its individual goal position $g^i$

$$R_{\text{pp}} = \sum_{i=1}^{N} - \max(\text{dist}(s^i, g^i), \delta), \tag{S4}$$

with $\delta = 0.025$. Similar to the PLAYGROUND-Push task, we use the $\delta$ value in the reward to ensure that the model doesn't over-optimize for each object. This is again set to be a more conservative distance threshold than the evaluation threshold $\delta_{\text{eval}} = 0.05$, that is set to be the same as in the original environment [24].

**CONSTRUCTION-Throwing**  The task is to throw blocks onto goal sites with size 0.2 by 0.2 (so 4 times the size of each block). Each goal site is at least 0.16 and at most 0.20 away from the manipulability range of the robot arm. For this task, we use sparse rewards together with a dense component. Throwing is a challenging task as (i) goal locations for objects are farther away, requiring a longer planning horizon and (ii) in the case of planning with dense rewards the agent can easily be stuck in a local optimum and push blocks outside of its manipulability range without actually reaching the goal location. In order to deal with (i), we keep a dense component for the reward and to address (ii), we also include sparse rewards and use a kernel for the dense reward component. For throwing, we only take the x-y positions of objects into account during the reward computation, such that our achieved block state is 2-dimensional with $s^i = [s_x^i, s_y^i] \in \mathbb{R}^D$. We evaluate the distance between each block position $s^i$ and the center location $g^i = [g_x^i, g_y^i]$ of each block's goal site individually across the x-y dimensions, such that the sparse component of the reward uses the evaluation:

$$R_{\text{sparse}}(s^i, g^i) = -1 + \prod_{d=1}^{D} [\![ |s_d^i - g_d^i| < \delta ]\!], \tag{S5}$$

where $\delta$ is 0.1, corresponding to the half-size of the goal site. The same value is used for $\delta_{\text{eval}}$ such that we don't require the whole block to be inside the goal site, but the block's center has to be inside the goal site for a successful throw. Note that we do not take the z-position of the object into account for the reward function. The dense reward component is given by:

$$R_{\text{dense}}(s^i, g^i) = \sum_{d=1}^{D} -1 + \exp\big(-0.5 \cdot |s_d^i - g_d^i|\big). \tag{S6}$$

For the overall throwing reward, we get the following function with $\eta = 0.001$:

$$R_{\text{throw}} = \sum_{i=1}^{N} R_{\text{sparse}}(s^i, g^i) + \eta \cdot R_{\text{dense}}(s^i, g^i). \tag{S7}$$

**CONSTRUCTION-Flipping**  The flipping task is defined as rotating the blocks $+90°$ around their x-axis. As in the original environment [24], the orientation information for each block is encoded in Euler angles in the state vector, where the xyz convention is used. As a result the angle values encode the relative rotating angles about x, y, and z axes in order, i.e. after we rotate about x, then we use the new (rotated) y, and the same for z. The flipping task reward thus only applies a constraint on the first Euler angle $\alpha_x$. We use sparse rewards for the flipping task. We also add a small dense component to the reward to incentivize the end effector to stay close to its initialization position $s^{\text{init}}$. We observed that this additional reward helps the robot find plans for flipping in-place as opposed to flicking objects from the side.

$$R_{\text{flip}} = \sum_{i=1}^{N} (-1 + [\![ \text{dist}(\alpha_x, 90°) < \delta ]\!]) - \eta \cdot \text{dist}(s^{\text{agent}}, s^{\text{init}}), \tag{S8}$$

with $\delta = 5°$ and $\eta = 0.001$.

### C.3.2 Planning Details for Downstream Tasks

We use slightly different controller settings for the different tasks as shown in Table S4. These parameters are shared between CEE-US and the unstructured baseline MLP + iCEM.

### C.3.3 Evaluation of Downstream Task Performance

In the PLAYGROUND environment, we evaluate the success rate of CEE-US and the unstructured baseline MLP + iCEM on the PLAYGROUND-Pushing task, when models taken from different training

**Table S4**: Settings for the iCEM controller used for zero-shot generalization in the extrinsic phase of CEE-US. Same settings were also used for the baseline MLP + iCEM. Any settings not specified here are the same as the general settings given in Table S3.

| Task | Controller Parameters | | | | |
|---|---|---|---|---|---|
| | Horizon $h$ | Colored-noise exponent $\beta$ | use_mean_actions | Noise strength $\sigma_{\text{init}}$ | Cost Along Trajectory |
| PLAYGROUND-Pushing | 40 | 3.5 | Yes | 0.8 | sum |
| CONSTRUCTION-Stacking | 30 | 3.5 | No | 0.5 | best |
| CONSTRUCTION-Pick & Place | 30 | 3.5 | Yes | 0.5 | best |
| CONSTRUCTION-Throwing | 35 | 2.0 | Yes | 0.5 | sum |
| CONSTRUCTION-Flipping | 30 | 3.5 | No | 0.5 | sum |

checkpoints are used for planning. Complementary to the results shown in Table 2 in the main text, Fig. S2 depicts the sample-efficiency of CEE-US compared to the unstructured baseline MLP + iCEM. In Fig. S1, we see that the learned GNN models' ability to capture object-object interactions leads to the selection of more efficient control plans like pushing two objects to the goal position at the same.

Table S5 and Table S6 contain the success rates reported for zero-shot generalization on downstream tasks in the CONSTRUCTION environment, complementary to Fig. 7 in the main text.

We spawn the environment with the number of objects specified in the table. The MLP + iCEM baseline that lacks combinatorial generalization, can only be applied to the 4 object case, as seen during the free-play phase. We do not perform masking of objects during planning for this baseline, as we consider a task with e.g. 2 object stacking to be defined in an environment spawned with the same amount of objects. For CONSTRUCTION-Stacking, in addition to the single-tower stacking, we also test for the multi-tower task with 4 objects (denoted by $2 + 2$) such that the goal is to build two towers with 2 blocks each. Since the base of these towers can be close to one another, this task has an increased level of difficulty compared to stacking 2 blocks, also reflected in the success rates shown in Table S5.

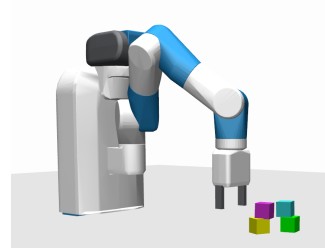

**Figure S3**: Multi-tower stacking task (Here task solved by CEE-US).

In the stacking tasks (single-tower and multi-tower), success is 1, only when the required towers with all the objects present in the environment are fully stacked. In all the other tasks in CONSTRUCTION and in PLAYGROUND-Pushing, the success rate in the multi-object setup is defined as the fraction of objects solved relative to the total number of objects spawned in the environment. For example, in an environment with 4 objects, 0.75 success rate means 3 out of 4 objects reached their respective goal positions.

### C.4   Hyperparameter settings for baselines

The hyperparameters for the model architecture and the training of the unstructured baseline MLP + iCEM, which corresponds to CEE-US without GNNs, are given in Table S7.



(a) $t = 0$       (b) $t = 100$       (c) $t = 300$

**Figure S1**: Downstream task PLAYGROUND-Pushing: move all objects to target location (red), as solved by CEE-US. At $t = 100$, we see that the planner can find a trajectory pushing two objects at the same time with the learned GNN models. At $t = 300$, the task is already solved.

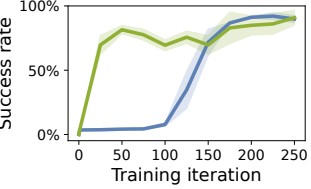

**Figure S2**: Success rate of downstream task in PLAYGROUND for CEE-US vs. MLP + iCEM evaluated at different training iterations.

**Table S5**: Zero-shot generalization performance of CEE-US vs. MLP + iCEM on downstream tasks CONSTRUCTION-Pick&Place and CONSTRUCTION-Stacking.

| Task | Pick&Place | | | | | Stacking | | | |
|---|---|---|---|---|---|---|---|---|---|
| # Objects | 2 | 3 | 4 | 5 | 6 | 2 | 3 | 4 | 2+2 |
| CEE-US | $0.94 \pm 0.01$ | $0.96 \pm 0.01$ | $0.97 \pm 0.01$ | $0.96 \pm 0.01$ | $0.96 \pm 0.001$ | $0.91 \pm 0.02$ | $0.27 \pm 0.08$ | $0.02 \pm 0.02$ | $0.236 \pm 0.06$ |
| MLP-iCEM | - | - | $0.83 \pm 0.02$ | - | - | - | - | $0 \pm 0$ | $0.013 \pm 0.012$ |

**Table S6**: Zero-shot generalization performance of CEE-US vs. MLP + iCEM on downstream tasks CONSTRUCTION-Throwing and CONSTRUCTION-Flipping.

| Task | Throwing | | | Flipping | | | | |
|---|---|---|---|---|---|---|---|---|
| # Objects | 2 | 3 | 4 | 2 | 3 | 4 | 5 | 6 |
| CEE-US | $0.675 \pm 0.039$ | $0.67 \pm 0.04$ | $0.61 \pm 0.01$ | $0.94 \pm 0.01$ | $0.93 \pm 0.03$ | $0.88 \pm 0.04$ | $0.87 \pm 0.04$ | $0.83 \pm 0.04$ |
| MLP-iCEM | - | - | $0.32 \pm 0.03$ | - | - | $0.25 \pm 0.02$ | - | - |

**Table S7**: Base settings for MLP model training in MLP + iCEM.

(a) General settings.

| Parameter | Value |
|---|---|
| Network Size | $3 \times 256$ |
| Activation function | SiLU |
| Ensemble Size | 5 |
| Optimizer | ADAM |
| Batch Size | 256 |
| Epochs | 50 |
| Learning Rate | 0.0001 |
| Weight decay | $5 \cdot 10^{-5}$ |
| Weight Initialization | Truncated Normal |
| Normalize Input | Yes |
| Normalize Output | Yes |
| Predict Delta | Yes |

(b) Environment-specific settings.

PLAYGROUND

| Parameter | Value |
|---|---|
| Network Size | $3 \times 128$ |
| Batch Size | 128 |

CONSTRUCTION

| Parameter | Value |
|---|---|
| Same as general settings | |

For the other baselines RND [17], Disagreement [16] and ICM [6], we use the implementation from Laskin et al. [25] that uses DDPG [26] with the same hyperparameter settings proposed there. The code for these baselines can be found in `https://github.com/rll-research/url_benchmark`.

### C.5 Offline RL

**Policy Selection** Since CQL tends to overfit to the training data, resulting in a significant drop in task performance, we use early stopping to select the best policy on a run by run basis.

**Rewards** To train the policies with offline RL, we use sparse rewards in all the experiments. The reward is computed according to:

$$r(g^{\text{achieved}}, g^{\text{desired}}) = [\![\text{dist}(g^{\text{achieved}}, g^{\text{desired}}) < \delta]\!] \in [0, 1], \tag{S9}$$

with $g^{\text{achieved}}$ being the achieved goal, $g^{\text{desired}}$ being the desired goal and $\delta$ being a task-dependent threshold. Depending on the task, $g^{\text{achieved}}$ is equal to $s^{\text{agent}}$ or $s^i$, $i = 1, \ldots, N$, or any combination of these.

**State Representation** In PLAYGROUND, we use the flat state representation provided by the environment as input for the CQL algorithm. In CONSTRUCTION, we also use the flat state representation provided by the environment, including the relative positions of the objects to the end-effector position. Without this relative information, CQL could not learn a policy for the object manipulation task.

**Datasets** We use the data collected by the different intrinsically motivated agents during free play as datasets for offline RL. The same amount of free-play data is used from the different agents to generate

**Table S8**: Settings for offline RL.

(a) Settings for CQL.

| Parameter | Value |
|---|---|
| Batch size | 256 |
| Actor learning rate | 1.0e-4 |
| Critic learning rate | 3.0e-4 |
| Temp learning rate | 1.0e-4 |
| Alpha learning rate | 0.0 |
| Conservative weight | 10.0 |
| Number of action samples | 10 |
| q_func_factory | mean |
| Optimizer | ADAM |
| Actor Encoder Network Size | $3 \times 256$ |
| Critic Encoder Network Size | $3 \times 256$ |
| gamma | 0.99 |
| tau | 0.005 |
| n_critics | 2 |
| Initial temperature | 1.0 |
| Initial $\alpha$ | 1.0 |
| $\alpha_{\text{threshold}}$ | 10.0 |
| Conservative weight | 1.0 |
| soft_q_backup | No |

(b) HER Settings

| PLAYGROUND | |
|---|---|
| **Parameter** | **Value** |
| Replay Strategy | Future |
| replay_k | 4 |

the datasets. In PLAYGROUND, the datasets contain $500,000$ transitions. In CONSTRUCTION, the datasets contain $600,000$ transitions.

**Tasks** In PLAYGROUND, we evaluate the performance of offline RL trained with the different datasets on two tasks: (i) move the agent to a randomly sampled target location and (ii) move the first object to a randomly sampled target location. In CONSTRUCTION, we evaluate CQL on (i) move the end-effector to a randomly sampled location and (ii) move the first object to a randomly sampled target location that is in the air in $50\%$ of the cases.

### C.6 Uncertainty Heatmaps

The uncertainty heatmaps are computed by a spatial discretization of the playground area (bin size equal to $r/10$, where $r$ is the agent's radius) and evaluating Eq. 6 for 8 actions, equidistant unit vectors on the unit circle, for the agent hypothetically being at each $(x, y)$ location of the grid. This means that we also spawn the agent inside objects and walls, violating the regular physical properties of the environment. As a result, there is always remaining uncertainty inside the walls and inside the objects.

The progression of the uncertainty heatmaps shown in Fig. 3 can be interpreted as follows:

- **Iteration 1**: After only one training iteration of CEE-US, we still have uniform uncertainty of the model as the model lacks training and the model's predictions are very inaccurate.

- **Iteration 25**: As more data of agent-object interactions are collected, the model discovers objects as a source of uncertainty, resulting in high epistemic uncertainty around the objects. As the models also start learning that the agent cannot permeate objects, i.e. agent and object cannot occupy the same space, the uncertainty at the center of objects is also high. Notice how this isn't the case for the variant MLP+iCEM (see Fig. 3). As the MLP models lack data from agent-object interactions, the model is confident (low ensemble disagreement) that the agent can simply move through objects. Same principle also applies to the walls. Before CEE-US generates enough agent-wall interactions, there is no reason for the agent to expect any different dynamics at the boundaries of the playground since the walls are not part of the state information. For instance, the GNN ensemble doesn't have enough data generated at the left wall at iteration 25, resulting in low ensemble disagreement.

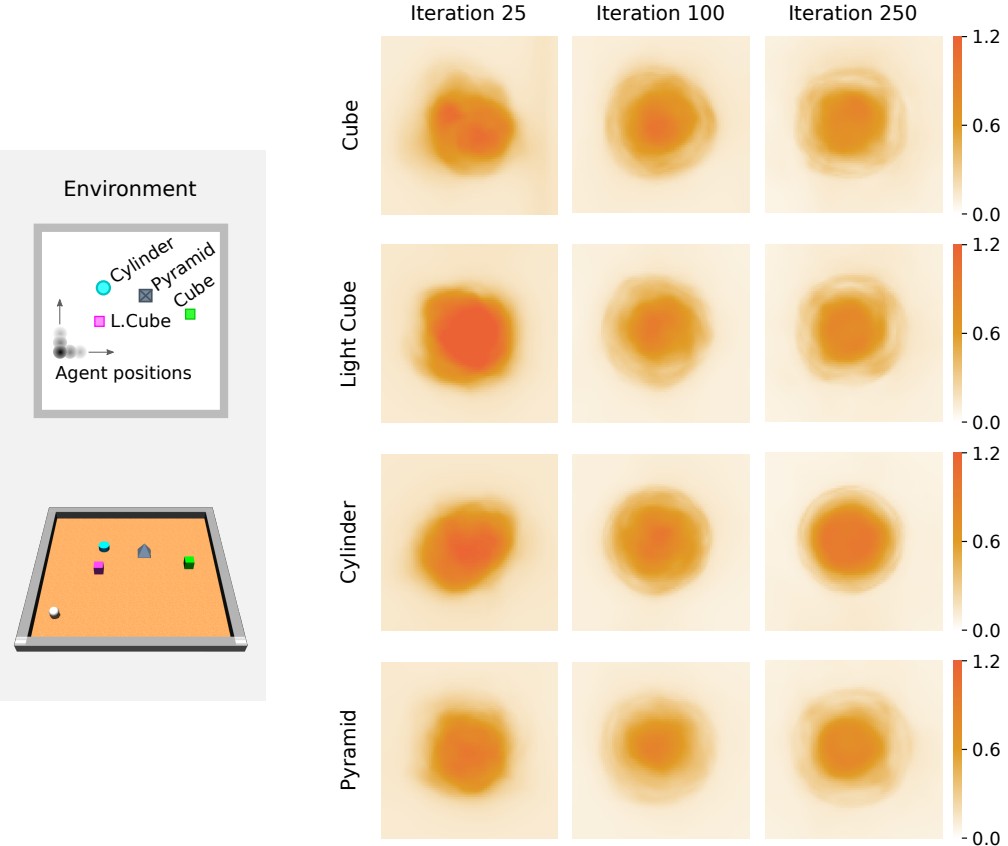

**Figure S4**: **Close-up of each object in uncertainty heatmaps of the GNN ensemble with CEE-US in PLAYGROUND.**

- **Iterations 100 - 249**: As CEE-US generates more agent-object and object-object interactions, the uncertainty around the object boundaries starts decreasing as the GNN ensemble learns more about each object's dynamics. As demonstrated in Fig. S4, the objects' shapes (cube, cylinder, pyramid) becomes discernible with more training iterations. As explained above, the uncertainty at the center of each object always remains.

We generate similar uncertainty heatmaps for the CONSTRUCTION environment as shown in Fig. S5. Here, we only put the robot gripper on different locations on the table, more precisely on a 80 cm × 80 cm square grid around the initial gripper position, that is discretized into bins of size 0.005 mm corresponding to $1/10$th of the cube size. In order to obtain the uncertainty heatmaps, we evaluate the ensemble disagreement Eq. 6 for 100 random actions at each hypothetical location of the robot arm in the spatially discretized grid on the table.

## D  Multi-step Prediction Performance of GNNs and MLPs

In the PLAYGROUND environment, we showcase the multi-step prediction performance of the trained GNN vs. MLP dynamics models at the end of free play. For a given starting state of the environment at $t = 0$ and an action sequence $(a_t)_{t=0}^{H-1}$, we generate a rollout in imagination of the trained models and compare these multi-step dynamics predictions to the ground truth. An example trajectory can be found in Fig. S6. We compute the cumulative prediction error of the generated trajectories, taking the mean predictions across the ensemble members for the GNNs and MLPs respectively, on 50 random evaluation rollouts with a multi-step prediction horizon of 50 timesteps. The evaluation rollouts are generated using a random policy, that interacts with one or more randomly chosen objects at each rollout. Using the GNN ensemble we get a prediction error of 2.82, whereas for the MLP ensemble we get 4.06 (cumulative error over the time horizon as well as the state space dimension).

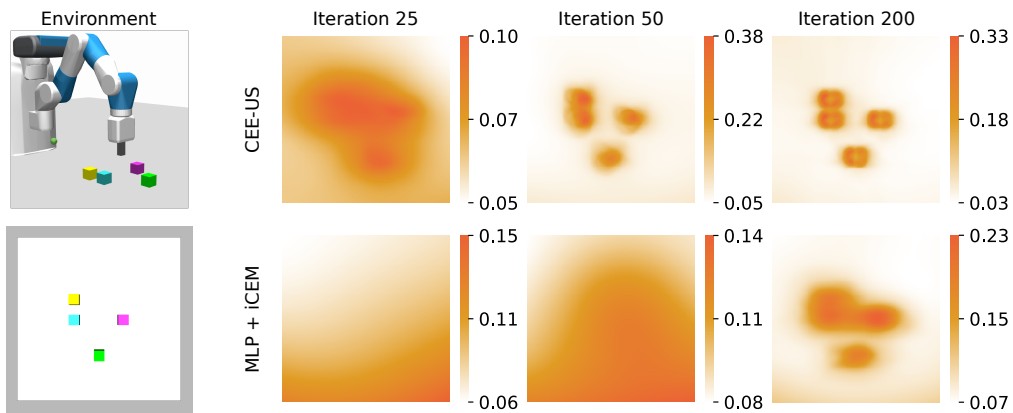

**Figure S5**: **Uncertainty heatmaps of the world model ensembles with CEE-US in CONSTRUC-TION, compared to the unstructured baseline MLP+iCEM.** Similar to the results shown in the PLAYGROUND environment, the uncertainty in CONSTRUCTION is also localized around the objects early on in the training, with object shapes becoming even more discernible further along in free play at iteration 200.

This illustrates the improved dynamics prediction that is obtained through the use of structured world models.

In Fig. S7, we also show the behavior of MLP+iCEM later on during free play in terms of interaction metrics, where we train it for an additional 100 iterations. In Fig. S8, we show the downstream task performance on the Pick & Place and Flipping tasks for models checkpointed at different iterations of free play. Even with an additional 100 iterations of free play, MLP+iCEM's downstream task performance is inferior to CEE-US. Overall, this showcases the importance of the accurate forward dynamics prediction of GNNs not just in terms of sample-efficiency of interaction metrics, but also for zero-shot downstream task generalization.

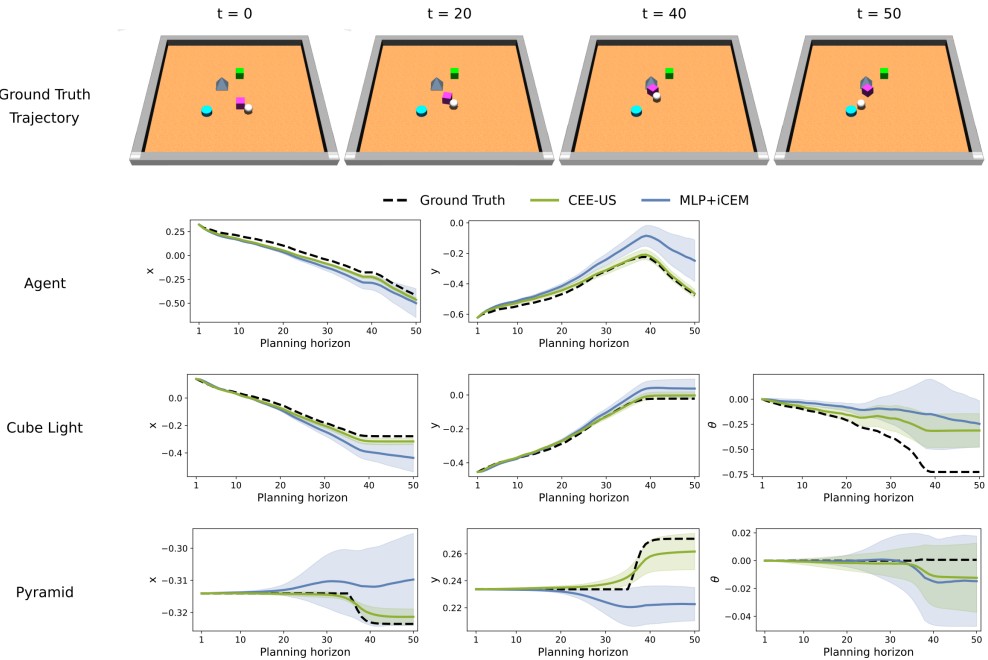

**Figure S6**: **Multi-step prediction performance of the trained GNN ensemble with CEE-US at the end of free play compared to the trained MLP ensemble with MLP+iCEM.**

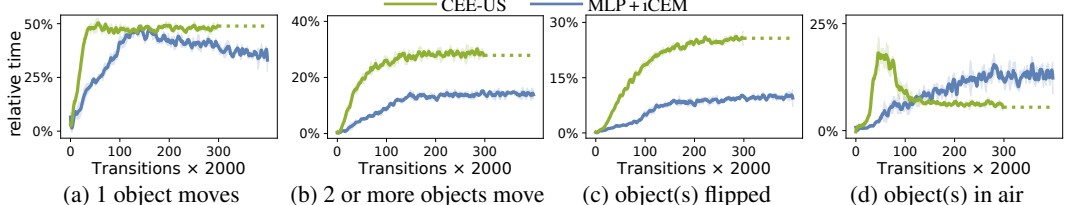

|  (a) 1 object moves | (b) 2 or more objects move | (c) object(s) flipped | (d) object(s) in air |

**Figure S7**: **Asymptotic behavior of MLP + iCEM during free play in CONSTRUCTION compared to CEE-US**. Interaction metrics of free-play exploration count the relative amount of time steps spent in moving one object (a), moving two and more objects (b), flipping object(s) (c), and moving objects into the air (d). This is the extension of Fig. 5 in the main paper. Here the baseline MLP + iCEM is trained for an additional 100 training iterations (corresponding to 200K more transitions collected). For CEE-US, we use the dashed line to visualize the achieved interaction metrics at the end of training at iteration 300. Three independent seeds were used.

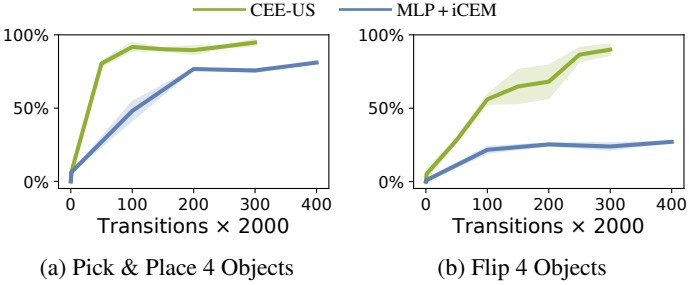

|  (a) Pick & Place 4 Objects | (b) Flip 4 Objects |

**Figure S8**: **Asymptotic downstream task performance of MLP + iCEM in CONSTRUCTION compared to CEE-US**. We compute the achieved success rates for models checkpointed over the course of free play, where we train the MLP + iCEM for an additional 100 training iterations (corresponding to 200K more data points collected). Three independent seeds were used.

# E   Combining Model-based Control with Random Network Distillation

One key element of our method is that we use the ensemble disagreement to approximate the epistemic uncertainty of the model itself. This is inherently different from the intrinsic rewards computed for instance in Random Network Distillation (RND) [17]. RND is essentially an expansion of count-based methods to continuous domains and the intrinsic reward is decoupled from the actual model performance of the dynamics model. Note that in the case of RND, as explained in Sec. 3.2, the RND module tries to match the output of a random target network. As long as a state is not visited enough, the RND module will generate high intrinsic reward regardless of whether the model can already predict this state accurately or not. In the opposite scenario, even if a state is trivial, the RND module is agnostic to the invariances and symmetries in an environment. As a result, it will try to create state-space coverage even when the state transition dynamics is trivial to learn. In order to test how using the RND intrinsic reward for planning affects behavior during free play, as well as the consequent downstream task performance, we ran new baselines that we refer to as GNN + RND and MLP + RND. In these baselines, we have a world model learning the dynamics (GNN for GNN + RND and an MLP for MLP + RND) which is used for model-based planning during free play. However, instead of using ensemble disagreement, we use the intrinsic reward of a separate RND module (MLP) to structure the free play. The RND network is also trained on the generated free-play data, separately from the actual world model.

Figure S9 illustrates that the GNN + RND and MLP + RND both produce less single- and multi-object interactions and flipping behavior than their ensemble disagreement counterparts. In terms of object(s) in air time, GNN + RND surpasses CEE-US. This is expected as once the GNN ensemble learns the lifting behavior, this knowledge is shared among all objects. The GNN ensemble focuses more on bringing two objects together, flipping and/or rolling them. In the case of RND, covering the whole air space with the different cubes is still incentivized since there is no connection between the dynamics model and the RND module. However, this behavior does not necessarily lead to superior task performance for the RND variant, as shown in Fig. S10. There is a large difference in

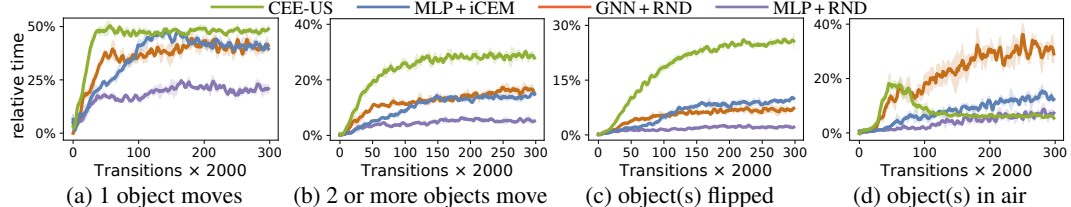

(a) 1 object moves    (b) 2 or more objects move    (c) object(s) flipped    (d) object(s) in air

**Figure S9**: **Interactions generated during free play in CONSTRUCTION by CEE-US**. Interaction metrics of free-play exploration count the relative amount of time steps spent in moving one object (a), moving two and more objects (b), flipping object(s) (c), and moving objects into the air (d). We compare CEE-US and MLP + iCEM, which use the epistemic uncertainty of the model approximated via the ensemble disagreement, with variants that use a separate RND module for intrinsic reward. We used three independent seeds.

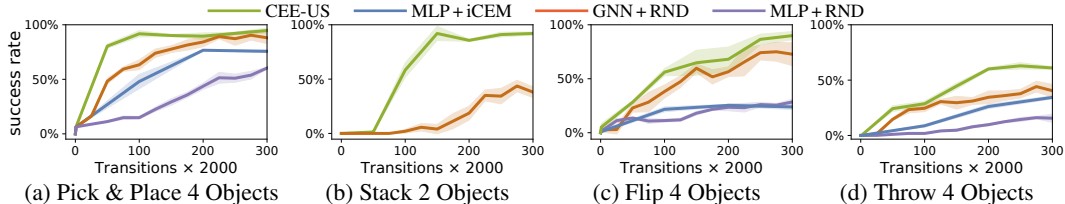

(a) Pick & Place 4 Objects    (b) Stack 2 Objects    (c) Flip 4 Objects    (d) Throw 4 Objects

**Figure S10**: **Downstream task performance in CONSTRUCTION by CEE-US and the baselines for models checkpointed over the course of free play**. We compare CEE-US and MLP + iCEM, which use the model's epistemic uncertainty approximated via the ensemble disagreement as intrinsic reward, with variants that use a separate RND module. We used three independent seeds.

the achieved success rates after 300 training iterations, where each training iteration corresponds to collecting 2000 Transitions and training the models. In the case of flipping, CEE-US is again superior to the GNN + RND variant. On the pick & place task, similar end performance is reached and yet CEE-US reaches better performance faster. Despite the fact that GNN + RND collects a lot of data with objects in air between training iterations 100-200, this doesn't culminate in any significantly better task performance in the pick & place and stacking tasks, where lifting is a key component. Similarly for the throwing task, we observe the superior performance of the ensemble disagreement-based methods, CEE-US and MLP + iCEM, over their RND counterparts.

These results showcase the sample-efficiency of our method not just in terms of generated interactions, but also in terms of zero-shot downstream task performance.

Another important observation in these experiments is the sample-efficiency we obtain through model-based planning alone. If we compare MLP + RND with the standard RND baseline performance that is trained with an exploration policy as shown in Fig. 5, we get much more interaction-rich exploration during free-play. This again highlights the importance of planning for multi-step intrinsic rewards into the future.

# F    Preliminary Results for CEE-US in ROBODESK

We apply CEE-US to the ROBODESK environment [53] (Fig. S11) in order to test if our method can deal with diverse geometries of objects with only proprioceptive state information. This environment has complex objects/entities such as a drawer, a sliding cabinet, buttons and other blocks.

**Table S9**: Preliminary success rates for zero-shot generalization in the extrinsic phase of CEE-US in ROBODESK.

|  | **Task** | | | |
| --- | --- | --- | --- | --- |
|  | Open Drawer | Open Slide Cabinet | Push Green Button | Push Flat Block Off Table |
| CEE-US | $0.97 \pm 0.02$ | $0.87 \pm 0.07$ | $0.87 \pm 0.07$ | $0.58 \pm 0.09$ |

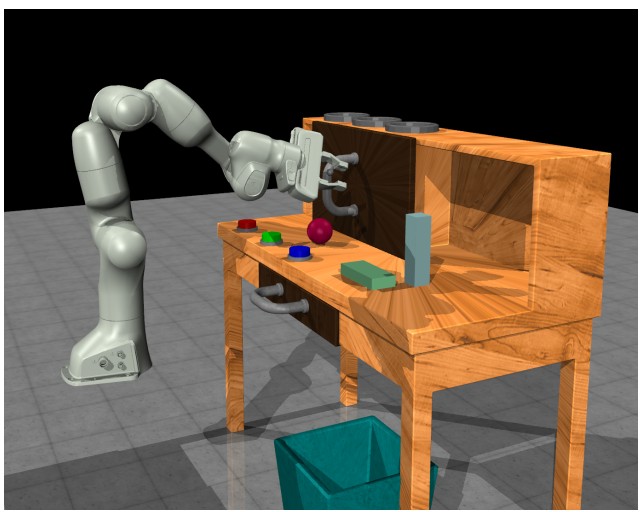

**Figure S11**: ROBODESK environment.

For ROBODESK, we encode each entity's state purely as proprioceptive information of position, quaternion, linear and angular velocities. Note that the entities have even different joint types, where the drawer has a slide/prismatic joint along y-axis, the sliding cabinet a slide joint only in x-axis, the buttons slide joints in z-axis, and the blocks and the ball corresponding to free joints. The different entity types are encoded as static object features and are categorical variables with one-hot encoding.

In our experiments, during the intrinsic phase of CEE-US, the robot arm interacts with the different entities, e.g. opening drawer and cabinet, pushing blocks and pushing buttons. The learned GNN ensemble can then be used in the extrinsic phase to solve downstream tasks zero-shot. We test opening the drawer, sliding the cabinet, pushing buttons, and moving blocks yielding the following success rates shown in Table S9.

The corresponding videos can be found on our supplementary website https://martius-lab.github.io/cee-us.