# OpenReview forum: "Curious Exploration via Structured World Models Yields Zero-Shot Object Manipulation"
_NeurIPS.cc/2022/Conference — NeurIPS 2022 Accept_

### Official Review · Reviewer_JJXz · 2022-07-10

**Rating:** 6
**Confidence:** 4
**Soundness:** 3 good
**Presentation:** 3 good
**Contribution:** 2 fair

**Summary:**

This paper proposes a curiosity-driven method for exploration via structured world models. It uses GNN ensemble disagreement for computing intrinsic motivation signals in RL, which achieves sample-efficient and interaction-rich exploration in multi-object manipulation environments. Further, the trained world models combined with planning-based methods can achieve zero-shot generalization to many object manipulation tasks. Experiments shows the data-efficiency, the zero-shot generalization ability and manipulation accuracy of the proposed method.



**Questions:**

1.	Could the author show in detail how to train the proposed method using image observations instead of proprioceptive states as the input, and further transfer the proposed method in real-world applications?
2.	Could the author show the heaviness (e.g., the training time of the ensemble of M GNNs compared the MLP baseline) of the proposed method? Also, I wonder if it is possible to use some designed intrinsic rewards instead of the ensemble of M GNNs to train a world model with good data-efficiency?


**Ethics Review Area:**

["I don’t know"]

**Limitations:**

The most significant limitation of this paper is the limited application in the real-world setting as the proprioceptive states instead of image observations used in the paper acts as an important constraint.
Besides, the heaviness in using an ensemble of GNNs may be a limitation. And this limitation may be even serious when the setting becomes more complex, which requires the ensemble to contain more GNNs with more model parameters.


**Strengths And Weaknesses:**

Strengths:
1.	This paper achieves quite a high sample-efficiency compared with other methods using the curiosity-driven method.
2.	This paper uses GNN with high combinatorial generalization abilities, which makes the method generalizes to environment with novel number of objects.
3.	The world models trained using the proposed method can easily generalize to many downstream manipulation tasks without specific trainings.


Weakness:
1.	The world model takes proprioceptive states of objects, instead of the image observation. This may be a hinder to the transfer to the real-world applications, in that: (1) we can not easily get the accurate proprioceptive states of objects; (2) objects with diverse geometries (e.g., real-world cabinets) do not have simple proprioceptive states, and therefore need image observations as input, otherwise the application range will be limited; (3) training using images has a much harder setting than proprioceptive states, which may potentially require the adjustment of the method proposed in this paper. Therefore, in my opinion, it is really important to train the proposed method based on image observations, instead of proprioceptive states.
2.	Though resulting in good data-efficiency, using an ensemble of M GNNs makes it heavy to train the model. Training with designed intrinsic rewards (e.g., the number of agent-object / object-object interactions, the state prediction accuracy, etc) may make the training simpler with only one GNN, while keeping the curiosity-driven good data-efficiency.

---

> ### Author Response · Authors · 2022-08-02
> **Response to Reviewer 3 [1/2]**
>
> Thank you for your time reviewing our work and your valuable feedback. We have improved our paper based on your concerns, as addressed in the following:
>
> > Could the author show in detail how to train the proposed method using image observations instead of proprioceptive states as the input, and further transfer the proposed method in real-world applications?
>
> We acknowledge that our method in its current version is only applied to proprioceptive inputs. However, as discussed in the Discussion part of our paper, our method can be extended in future work to deal with images by combining CEE-US with object-centric unsupervised representation learning methods, which exactly produce the state representations we use, namely the location and visual properties of each object/entity separately. One such example is the method SCALOR [1]. Below we provide more information about these methods.
> Combining SCALOR with our pipeline in its current form requires some non-trivial additional technical effort, such that we defer this to future work. However, we believe that it is well possible to make CEE-US deal with image input in this way.
>
> In relation to applicability to images, you raised another concern:
>
> > objects with diverse geometries (e.g., real-world cabinets) do not have simple proprioceptive states, and therefore need image observations as input, otherwise the application range will be limited
>
> In order to test if our method can deal with diverse geometries of objects with only proprioceptive state information, we now ran experiments on the Robodesk environment (https://github.com/google-research/robodesk). This environment has complex objects/entities such as a drawer, a sliding cabinet, buttons and other blocks. In our experiments, we observe the robot arm interacting with the entities during free play, e.g. opening drawer and cabinet, pushing blocks and pushing buttons. The corresponding videos can be found on our supplementary website: https://cee-us.github.io.
>
> For Robodesk, we encode each entity’s state purely as proprioceptive information of position, quaternion, linear and angular velocities. Note that the entities have even different joint types, where the drawer has a slide joint in only $y$ direction and the sliding cabinet a slide joint only in $x$ direction, and the blocks and the ball being free joints. The different entity types are encoded as static object features and are categorical variables with one-hot encoding. An easy way of obtaining this information in real-world setups (where we do not have access to the ground-truth states) is by using motion capturing for object tracking as it is commonly done in robotics.
>
> The learned GNN ensemble can then be used in the extrinsic phase to solve downstream tasks zero-shot. We test opening the drawer, sliding the cabinet, pushing buttons, and moving blocks yielding the following success rates:
>
> | Open Drawer | Open Slide Cabinet | Push Green Button | Push Flat Block Off Table |
> |-------------|--------------------|-------------------|---------------------------|
> | 0.97 ± 0.02 | 0.873 ± 0.066      | 0.870 ± 0.071     | 0.58 ± 0.09               |
>
> We agree that image features are an important part of perception and aid learning correct dynamics. However, our results in the Robodesk environment show that CEE-US with proprioceptive inputs can still be successfully applied to a more realistic environment.

---

> ### Comment · Reviewer_JJXz · 2022-08-05
> **Thanks for authors' response**
>
> The rebuttal from authors have addressed my concerns in:
> (1) how to train the proposed method using image observations
> (2) why this method is designed to use ensemble of GNNs and the corresponding heaviness.
> Therefore, I have re-rated the evaluation to 6 (weak accept).

---

### Official Review · Reviewer_zPKR · 2022-07-11

**Rating:** 7
**Confidence:** 4
**Soundness:** 4 excellent
**Presentation:** 4 excellent
**Contribution:** 3 good

**Summary:**

This paper an agent that learns a world model through exploratory play, which it then deploys to solve downstream tasks via a planner. The world model is an ensemble of GNNs, where nodes represent objects, which introduces an object-centric inductive bias. Effective playful exploration is achieved by using disagreement among the ensemble as a proxy for uncertainty, which allows the agent to select actions whose outcomes it cannot yet reliably predict. The authors show that this method works better than baselines, in the sense that it quickly starts engaging in “interesting” interactions (with objects). They do this in two simulated environments with dynamics - a planar world with a variety objects that the agent can push around, and a robot-arm environment where the agent can stack, flip, and throw objects. The authors show that the resulting agent achieves zero-shot generalisation on downstream tasks, to numbers of objects unseen at training time.

**Questions:**

See below

**Limitations:**

I believe the ensemble disagreement method that the authors use is susceptible to the white noise problem, wherein the agent is motivated to fixate on dynamics that are random and therefore cannot be learned. Perhaps the authors could comment on this.

**Strengths And Weaknesses:**

The paper makes progress on an important problem in the RL field, namely learning with intrinsic motivation. As the authors argue, a learned world model that is reward-neutral is a key ingredient for better generalisation and transfer learning. The benefits of learning an object-centric representation in this context are well supported by the authors’ work. The experimental methodology seems sound, and the results are persuasive. The paper is well written and clear and, as far as I can determine, technically sound. Overall I think this is a very good paper that is well above the acceptance threshold for NeurIPS.

I am not sufficiently familiar with the literature on intrinsic motivation to assess the related work section, or to judge the originality of the work with great confidence (hence my “low confidence” rating). However, the following paper got my attention a few years back, which seems very related, and isn’t cited:

Nick Haber, Damian Mrowca, Li Fei-Fei, Daniel L. K. Yamins. Emergence of Structured Behaviors from Curiosity-Based Intrinsic Motivation. In Proceedings of the 40th Annual Meeting of the Cognitive Science Society, CogSci 2018. (https://arxiv.org/abs/1802.07461)

I notice that a subset of the authors of that paper have a more recent ICML paper that also looks relevant, and isn't in the bibliography: https://arxiv.org/abs/2007.07853

---

> ### Author Response · Authors · 2022-08-02
> **Response to Reviewer 2**
>
> We thank the reviewer for their review and for pointing us in the direction of the two mentioned papers. They are indeed closely related to our work, and we updated our manuscript to include these citations as following:
>
> We added Kim et. al, 2020 [1] to the list of prediction error based methods (l. 247) as states are retrospectively labeled as interesting based on the $\gamma$-Progress curiosity which is defined as the predicticion error between the current world model and an exponential average of old world models.
>
> We will add the following discussion to the final version of the paper (due to space constraints we could not include it now):
>
> _Haber et al. [2] implements directed information-seeking behavior by using a loss model to predict the prediction error at future time steps. However, the agent might get stuck on states that are notoriously difficult to predict due to e.g. stochasticity._
>
> Regarding the reviewer’s question on the white noise problem: this is indeed an important problem in novelty-based curiosity methods. However, deterministic ensembles inherently sidestep this problem as also put forward in Pathak et. al, 2019 [3]. Since the intrinsic reward in our case is computed as the disagreement of the ensemble members, the intrinsic reward goes to zero, when all members agree such that the variance of the predictions is zero. In the case of noise, with enough samples, each ensemble member should converge to predict the mean of the noise. This in turn means that the variance goes to zero and so the intrinsic reward also goes to zero. As a result, the agent shouldn’t get stuck in these situations. However, a more elegant way of dealing with inherent stochasticity would be to train stochastic ensembles outputting the parameters e.g. of a Gaussian distribution. This would help us isolate the aleatoric uncertainty from the epistemic uncertainty. This could potentially reduce the time spent in a noisy part of the state space, and learning the inherent system noise can also be exploited for risk-averseness later on in planning, as in [4].
>
> Since in our work we focused on deterministic environments, we didn’t include aleatoric uncertainty modeling. However, we see this as an exciting future direction.
>
> [1] Kuno Kim, Megumi Sano, Julian De Freitas, Nick Haber and Daniel Yamins. “Active World Model Learning with Progress Curiosity.” International Conference on Machine Learning. ICML, 2020.
>
> [2] Nick Haber, Damian Mrowca, Li Fei-Fei and Daniel L. K. Yamins. “Emergence of Structured Behaviors from Curiosity-Based Intrinsic Motivation.” Annual Meeting of the Cognitive Science Society. CogSci, 2018.
>
> [3] Pathak, Deepak, Dhiraj Gandhi, and Abhinav Gupta. "Self-supervised exploration via disagreement." International conference on machine learning. PMLR, 2019.
>
> [4] Vlastelica, Marin, et al. "Risk-averse zero-order trajectory optimization." 5th Annual Conference on Robot Learning. 2021.

---

> > ### Comment · Reviewer_zPKR · 2022-08-05
> > **Thanks for response**
> >
> > Many thanks to the authors for their response, and especially for clarifying the white noise issue. I remain impressed by the paper, and I'm now more confident in that assessment, so I have increased my confidence accordingly

---

### Official Review · Reviewer_Fu6z · 2022-07-11

**Rating:** 8
**Confidence:** 4
**Soundness:** 3 good
**Presentation:** 4 excellent
**Contribution:** 3 good

**Summary:**

The authors propose to leverage an ensemble of GNN-based forward-models to estimate an agent's uncertainty with interacting with some part of an object-centric state-space. This uncertainty is used as an intrinsic reward signal during task-agnostic environment exploration with a model (dubbed "Free Play"). Given a fully observable MDP, the authors show that this kind of "Free Play" enables qualitatively object-centric exploration along with the ability to perform zero-shot generalization to reward functions defined over object attributes.

**Questions:**

Experiments:
- Is is possible to run MLP + iCEM but instead of concatenating the object-representations, you simply sum them, or do something that exploits structure but is not a GNN? This would help isolate the benefits of using a GNN.

Questions:
- During evaluation, you give the task reward function to a planner in order to select the appropriate action. Your model however, only learns to predict the next state, given an action. It doesn't learn to predict the environment reward---in fact, the agent never sees environment reward. How does the planner know how to leverage the task reward to construct a plan towards that reward? Does the planner have knowledge about all ground-truth states, along with the reward R(s,a) for all ground-truth states?
- It seems that an MLP leads to the same relative amount of exploration with different objects as the GNN but takes more time to do so? Why do you think it leads to similar results?
- Figure 3: It seems structured world models lead to object interactions *more quickly*? With infinite time, how do you think of GNN & MLP would have the same asymptotic behavior?
- What are the numbers in Table 3? Success rate?

Baselines:
- Plan2Explore seems like a good baseline since it seems similar in principal but doesn't offer combinatorial optimization. Why is this not a baseline? Could you expand on the differences to this baseline?



**Limitations:**

Limitations are adequately addressed.

**Strengths And Weaknesses:**

Strengths:
* Very clear method, results, and introduction
* High potential for impact in psychology and robotics
* While the results are somewhat expected, no one has combined disagreement with GNN based methods and the results are convincing


Weaknesses:
* The method has some hand-crafting in parsing out the agent from the objects, though this seems like a fine first-step.

---

> ### Author Response · Authors · 2022-08-02
> **Response to Reviewer 1**
>
> We thank the reviewer for their review and for sharing our excitement on this work’s potential impact at the intersection of robotics and psychology.
>
> > During evaluation, you give the task reward function to a planner in order to select the appropriate action.
>
> The world model only learns the dynamics of the environment, in a completely task-agnostic fashion. We take the stance that a downstream task is given with a reward function that the agent can evaluate on real and imagined states. (Typically, either a distance to a desired goal, or a sparse reward). In future work, such a reward function could potentially be synthesized from a language instruction or similar. We give the planner access to this reward function, such that the cost of the state trajectories that were rolled out in the imagination of the learned GNN models can be evaluated. We would like to highlight that this is in line with other works in model predictive control, in which the planner has access to the cost/reward function. However, the planner doesn’t have any additional knowledge about the ground truth states in the environment, and only computes the reward/cost given the state trajectories produced in imagination of the model. If the model is wrong, then an actually high-reward action sequence can be mislabeled as low-reward. Therefore, accurate dynamics prediction of the learned world model is key here.
>
> > Is is possible to run MLP + iCEM but instead of concatenating the object-representations, you simply sum them, or do something that exploits structure but is not a GNN? This would help isolate the benefits of using a GNN.
>
> As we need to do dynamics predictions per node/object, we cannot put the aggregated information through a flat layer without a subsequent per object output head. It is therefore unclear to us, how we could put the structural bias only partially in.
> The important part for sample-efficiency is sharing the parameters of the node and edge update functions across the nodes in the GNN, as the experience gathered on one object/entity can generalize over the other objects. It would be indeed very interesting to investigate different variations of GNNs in future work.
>
> > It seems that an MLP leads to the same relative amount of exploration with different objects as the GNN but takes more time to do so? Why do you think it leads to similar results?
> > Figure 3: It seems structured world models lead to object interactions more quickly? With infinite time, how do you think of GNN & MLP would have the same asymptotic behavior?
>
> Structured world models indeed lead to object interactions more quickly due to the reasons discussed in the second paragraph to the previous question, resulting in high gains in sample-efficiency. In the case of the Playground environment, the asymptotic behavior of both methods are on par in terms of interactions and success rates. However, in terms of multi-step prediction accuracy of the dynamics, the GNNs are superior to MLPs. We have now improved the manuscript and added a new section to the supplementary (see section D) to illustrate this. We have also trained the MLP+iCEM variant for an additional 100 iterations in the more complex Construction environment and added this new figure (Fig. S6) to the supplementary. We see that the asymptotic behavior of both methods in the Construction environment show a discrepancy, especially for more complex behaviors such as two object interactions and flipping. And we also show in the newly added figure S7, that the asymptotic downstream task performance generalization of the MLP variant, even after an additional 100 steps of training is inferior to CEE-US.
> For Playground, the interaction metrics being similar is also due to the fact that the search space in the Playground environment is constrained by the walls, and as the objects have only 3 degrees of freedom (x-y translation and rotation around z axis), the complexity of the behavior that can be shown in free play is limited.
>
> > What are the numbers in Table 3? Success rate?
>
> These are the success rates for offline RL. We have now updated the caption in the main paper for clarity.
>
> > Plan2Explore seems like a good baseline since it seems similar in principal but doesn't offer combinatorial optimization. Why is this not a baseline? Could you expand on the differences to this baseline?
>
> We will add the following discussion to the final version of the paper (due to space constraints we could not include it now):
>
> _Plan2Explore works with the latent dynamics prediction model, that has a recurrent structure, and has been applied to domains with image observations. Since we are considering a fully-observable setting with proprioceptive states, we didn’t apply this image-based method as a baseline._

---

> > ### Comment · Reviewer_Fu6z · 2022-08-08
> > **Thanks for the clarification.**
> >
> > Thanks for the clarification.
> >
> > This was a nice paper and my score remains very positive.

---

### Author Response · Authors · 2022-08-02
**General Response**

We would like to thank all reviewers for their valuable feedback. Here we summarize the major changes we made to the manuscript based on the feedback we received from the reviewers:

- We added section D in the supplementary showing the multi-step prediction performance of GNN vs. MLP ensembles as well as the asymptotic behavior comparison. **(R1)**

- We added new citations to related work. **(R2)**

- We added Section E to the supplementary to compare our method to a new baseline that uses model-based control (with GNN and MLP) together with Random Network Distillation. **(R3)**

- We updated our supplementary website (https://cee-us.github.io) to contain the application of CEE-US to the Robodesk environment as proof of concept for applicability to more complex environments. **(R3)**

We also corrected minor typos in the main text and the supplementary.

---

### Meta-Review · Area_Chair_V5ys · 2022-08-28

**Recommendation:** Accept
**Confidence:** Certain

**Metareview:**

Reviewer zPKR summarizes the paper well: "This paper an agent that learns a world model through exploratory play, which it then deploys to solve downstream tasks via a planner. The world model is an ensemble of GNNs, where nodes represent objects, which introduces an object-centric inductive bias. Effective playful exploration is achieved by using disagreement among the ensemble as a proxy for uncertainty, which allows the agent to select actions whose outcomes it cannot yet reliably predict. The authors show that this method works better than baselines, in the sense that it quickly starts engaging in “interesting” interactions (with objects)."

All reviewers unanimously vote to accept the paper. The limitations of the work are well summarized in the comment: "The most significant limitation of this paper is the limited application in the real-world setting as the proprioceptive states instead of image observations used in the paper acts as an important constraint. Besides, the heaviness in using an ensemble of GNNs may be a limitation."

The benefits of the method outweigh the limitations. I hope authors work on extending their work to operate on sensory observations in the future.

**Award:**

No

---

### Decision · Program_Chairs · 2022-09-14

Accept